# Neural interactions in the human frontal cortex dissociate reward and punishment learning

Etienne Combrisson[1]*, Ruggero Basanisi[1], Maelle CM Gueguen[2], Sylvain Rheims[3], Philippe Kahane[4], Julien Bastin[2], Andrea Brovelli[1]*

[1]Institut de Neurosciences de La Timone, UMR 7289, CNRS, Aix-Marseille Université, Marseille, France; [2]Univ. Grenoble Alpes, Inserm, U1216, Grenoble Institut Neurosciences, Grenoble, France; [3]Department of Functional Neurology and Epileptology, Hospices Civils de Lyon and University of Lyon, Lyon, France; [4]Univ. Grenoble Alpes, Inserm, U1216, CHU Grenoble Alpes, Grenoble Institut Neurosciences, Grenoble, France

**\*For correspondence:**
e.combrisson@gmail.com (EC);
andrea.brovelli@univ-amu.fr (AB)

**Competing interest:** The authors declare that no competing interests exist.

**Abstract** How human prefrontal and insular regions interact while maximizing rewards and minimizing punishments is unknown. Capitalizing on human intracranial recordings, we demonstrate that the functional specificity toward reward or punishment learning is better disentangled by interactions compared to local representations. Prefrontal and insular cortices display non-selective neural populations to rewards and punishments. Non-selective responses, however, give rise to context-specific interareal interactions. We identify a reward subsystem with redundant interactions between the orbitofrontal and ventromedial prefrontal cortices, with a driving role of the latter. In addition, we find a punishment subsystem with redundant interactions between the insular and dorsolateral cortices, with a driving role of the insula. Finally, switching between reward and punishment learning is mediated by synergistic interactions between the two subsystems. These results provide a unifying explanation of distributed cortical representations and interactions supporting reward and punishment learning.

## eLife assessment

This is an **important** information-theoretic re-analysis of human intracranial recordings during reward and punishment learning. It provides **convincing** evidence that reward and punishment learning is represented in overlapping regions of the brain while relying on specific inter-regional interactions. This preprint will be interesting to researchers in systems and cognitive neuroscience.

## Introduction

Reward and punishment learning are two key facets of human and animal behavior, because they grant successful adaptation to changes in the environment and avoidance of potential harm. These learning abilities are formalized by the law of effect (*Thorndike, 1898*; *Bouton, 2007*) and they pertain the goal-directed system, which supports the acquisition of action-outcome contingencies and the selection of actions according to expected outcomes, as well as current goal and motivational state (*Dickinson and Balleine, 1994*; *Balleine and Dickinson, 1998*; *Balleine and O'Doherty, 2010*; *Dolan and Dayan, 2013*; *Balleine, 2019*).

At the neural level, the first hypothesis suggests that these abilities are supported by distinct frontal areas (*Pessiglione and Delgado, 2015*; *Palminteri and Pessiglione, 2017*). Indeed, an anatomical

dissociation between neural correlates of reward and punishment prediction error (PE) signals has been observed. PE signals are formalized by associative models (*Rescorla et al., 1972*) and reinforcement learning theory (*Sutton and Barto, 2018*) as the difference between actual and expected action outcomes. Reward prediction error (RPE) signals have been observed in the midbrain, ventral striatum and ventromedial prefrontal cortex (vmPFC) (*Schultz et al., 1997*; *O'Doherty et al., 2004*; *O'Doherty et al., 2001*; *Pessiglione et al., 2006*; *D'Ardenne et al., 2008*; *Steinberg et al., 2013*; *Palminteri et al., 2015*; *Gueguen et al., 2021*). Punishment prediction error (PPE) signals have been found in the anterior insula (aINS), dorsolateral prefrontal cortex (dlPFC), lateral orbitofrontal cortex (lOFC), and amygdala (*O'Doherty et al., 2001*; *Seymour et al., 2005*; *Pessiglione et al., 2006*; *Yacubian et al., 2006*; *Gueguen et al., 2021*). Evidence from pharmacological manipulations and lesion studies also indicates that reward and punishment learning can be selectively affected (*Frank et al., 2004*; *Bódi et al., 2009*; *Palminteri et al., 2009*; *Palminteri et al., 2012*). Complementary evidence, however, suggests that reward and punishment learning may instead share common neural substrates. Indeed, hubs of the reward circuit, such as the midbrain dopamine systems and vmPFC, contain neural populations encoding also punishments (*Tom et al., 2007*; *Matsumoto and Hikosaka, 2009*; *Plassmann et al., 2010*; *Monosov and Hikosaka, 2012*). Taken together, it is still unclear whether reward and punishment learning recruit complementary cortical circuits and whether differential interactions between frontal regions support the encoding of RPE and PPE.

To address this issue, we repose on recent literature proposing that learning reflects a network phenomenon emerging from neural interactions distributed over cortical-subcortical circuits (*Bassett and Mattar, 2017*; *Hunt and Hayden, 2017*; *Averbeck and Murray, 2020*; *Averbeck and O'Doherty, 2022*). Indeed, cognitive functions emerge from the dynamic coordination over large-scale and hierarchically organized networks (*Varela et al., 2001*; *Bressler and Menon, 2010*; *Reid et al., 2019*; *Panzeri et al., 2022*; *Thiebaut de Schotten and Forkel, 2022*; *Miller et al., 2024*; *Noble et al., 2024*) and accumulating evidence supports that information about task variables is widely distributed across brain circuits, rather than anatomically localized (*Parras et al., 2017*; *Saleem et al., 2018*; *Steinmetz et al., 2019*; *Urai et al., 2022*; *Voitov and Mrsic-Flogel, 2022*).

Accordingly, we investigated whether reward and punishment learning arise from complementary cortico-cortical functional interactions, defined as statistical relationships between the activity of different cortical regions (*Panzeri et al., 2022*), within and/or between brain regions of the frontal cortex. In particular, we investigated whether reward and punishment prediction errors are encoded by redundancy- and/or synergy-dominated functional interactions in the frontal cortex. The search for synergy- and redundancy-dominated interactions is motivated by recent hypotheses suggesting that a trade-off between redundancy for robust sensory and motor functions and synergistic interaction may be important for flexible higher cognition (*Luppi et al., 2024*). On one hand, we reasoned that redundancy-dominated brain networks may be associated with neural interactions subserving similar functions. Redundant interactions may appear in collective states dominated by oscillatory synchronization (*Engel et al., 2001*; *Varela et al., 2001*; *Buzsáki and Draguhn, 2004*; *Fries, 2015*) or resonance phenomena (*Vinck et al., 2023*). Such collective states may give rise to selective patterns of information flow (*Buehlmann and Deco, 2010*; *Kirst et al., 2016*; *Battaglia and Brovelli, 2020*). On the other, synergy-dominated brain networks may be associated with functionally-complementary interactions. Indeed, synergistic interactions have been reported between distant transmodal regions during high-level cognition (*Luppi et al., 2022*) and, at the microscale, in populations of neurons within a cortical column of the visual cortex and across areas of the visuomotor network (*Nigam et al., 2019*; *Varley et al., 2023*). The notion of redundant and synergistic interactions resonates with the hypothesis that brain interactions regulate segregation and integration processes to support cognitive functions (*Wang et al., 2021*; *Deco et al., 2015*; *Sporns, 2013*; *Finc et al., 2020*; *Cohen and D'Esposito, 2016*; *Braun et al., 2015*; *Shine et al., 2016*).

In order to study redundancy- and synergy-dominated interactions, we used formal definitions from Partial Information Decomposition (PID; *Williams and Beer, 2010*; *Wibral et al., 2017*; *Lizier et al., 2018*). The PID decomposes the total information that a set of source variables (i.e. pairs of brain signals) encodes about a specific target variable (i.e. prediction errors) into components representing shared (redundant) encoding between the variables, unique encoding by some of the variables, or synergistic encoding in the combination of different variables. Within this framework, we used a metric known as interaction information (*McGill, 1954*; *Ince et al., 2017*), which

quantifies whether a three-variable interaction (i.e. pairs of brain regions and the PE variable) is either synergy- or redundancy-dominated. We predicted that redundancy-dominated functional interactions would engage areas with similar functional properties (e.g. those encoding RPE), whereas synergy-dominated relations would be observed between areas performing complementary functions (e.g. the encoding of RPE and PPE).

We investigated neural interactions within and between four cortical regions, namely the aINS, dlPFC, lOFC, and vmPFC, by means of intracerebral EEG (iEEG) data collected from epileptic patients while performing a reinforcement learning task (*Gueguen et al., 2021*). We found various proportions of intracranial recordings encoding uniquely RPE or PPE signals or both, suggesting a local mixed representation of PEs. We then identified two distinct learning-related subsystems dominated by redundant interactions. A first subsystem with RPE-only interactions between the vmPFC and lOFC, and a second subsystem with PPE-only interactions between the aINS and dlPFC. Within each redundant-dominated subsystem, we demonstrated differential patterns of directional interactions, with the vmPFC and aINS playing a driving role in the reward and punishment learning circuits, respectively. Finally, these two subsystems interacted during the encoding of PE signals irrespectively of the context (reward or punishment), through synergistic collaboration between the dlPFC and vmPFC. We concluded that the functional specificity toward reward or punishment learning is better disentangled by interactions compared to local representations. Overall, our results provide a unifying explanation of distributed cortical representations and interactions supporting reward and punishment learning.

## Results
### iEEG data, behavioral task, and computational modeling
We analyzed iEEG data from sixteen pharmacoresistant epileptic patients implanted with intracranial electrodes (*Gueguen et al., 2021*). A total of 248 iEEG bipolar derivations located in the aINS, dlPFC, vmPFC, and lOFC regions (*Figure 1A*) and 1788 pairs of iEEG signals, both within and across brain regions (*Figure 1B*) were selected for further analysis. Single subject anatomical repartition is shown in *Figure 1—figure supplement 1*. Participants performed a probabilistic instrumental learning task and had to choose between two cues to either maximize monetary gains (for reward cues) or minimize monetary losses (for punishment cues) (*Figure 1C*). Overall, they selected more monetary gains and avoided monetary losses but the task structure was designed so that the number of trials was balanced between reward and punishment conditions (*Figure 1D*).

We estimated trial-wise prediction errors by fitting a Q-learning model to behavioral data. Fitting the model consisted in adjusting the constant parameters to maximize the likelihood of observed choices. We used three constant parameters: (i) the learning rate α accounting for how fast participants learned new pairs of cues; (ii) the choice temperature β to model different levels of exploration and exploitation; (iii) θ parameter to account for the tendency to repeat the choice made in the previous trial. The RPE and PPE were obtained by taking the PE for rewarding and punishing pairs of cues, respectively. RPE and PPE showed high absolute values early during learning and tended toward zero as participants learned to predict the outcome (*Figure 1E*). The convergence toward zero of RPE and PPE was stable at the single subject level (*Figure 1—figure supplement 2*).

### Local mixed encoding of PE signals
At the neural level, we first investigated local correlates of prediction error signals by studying whether RPEs and PPEs are differentially encoded in prefrontal and insular regions. To this end, we performed model-based information theoretical analyses of iEEG gamma activities by computing the mutual information (MI) between the across-trials modulations in RPE or PPE signals and the gamma band power in the aINS, dlPFC, lOFC and vmPFC. The MI allowed us to detect both linear and non-linear relationships between the gamma activity and the PE. Preliminary spectrally-resolved analyses showed that the frequency range significantly encoding prediction errors was between 50 and 100 Hz (*Figure 2—figure supplement 1*). We thus extracted for each trial time-resolved gamma power within the 50–100 Hz range using a multi-taper approach for further analyses. MI analysis between gamma power and prediction error signals displayed significant group-level effects in all four cortical regions (*Figure 2A*) and globally reproduced previous findings based on general linear model analyses (*Gueguen et al., 2021*). Interestingly, we observed a clear spatial dissociation between reward

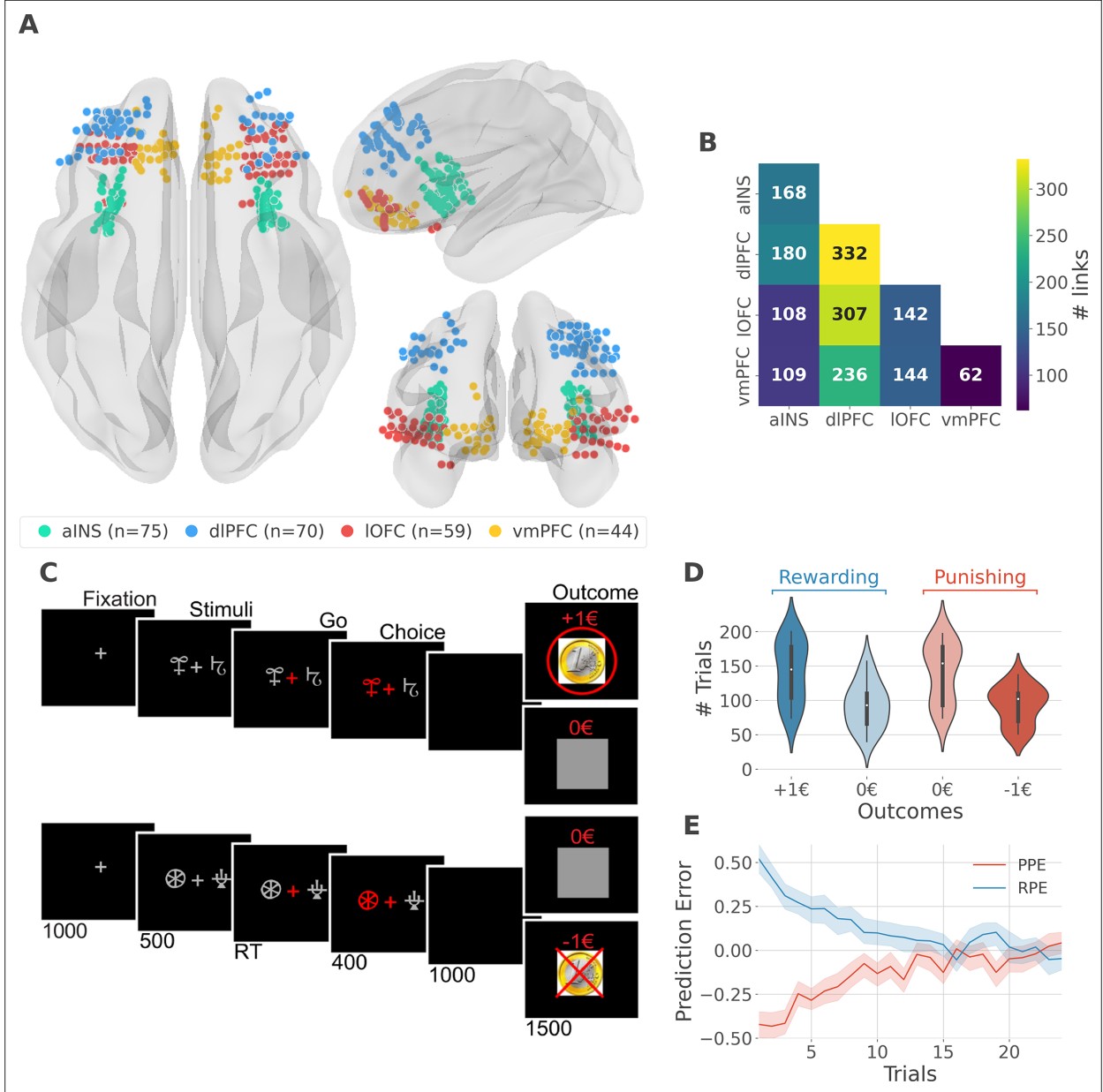

**Figure 1.** intracerebral EEG (iEEG) implantation, behavioral task, and computational modeling. (**A**) Anatomical location of intracerebral electrodes across the 16 epileptic patients. Anterior insula (aINS, n=75), dorsolateral prefrontal cortex (dlPFC, n=70), lateral orbitofrontal cortex (lOFC, n=59), ventromedial prefrontal cortex (vmPFC, n=44), (**B**) Number of pairwise connectivity links (i.e. within patients) within and across regions, (**C**) Example of a typical trial in the reward (top) and punishment (bottom) conditions. Participants had to select one abstract visual cue among the two presented on each side of a central visual fixation cross and subsequently observed the outcome. Duration is given in milliseconds, (**D**) Number of trials where participants received outcomes +1€ (142±44, mean ± std) vs. 0€ (93±33) in the rewarding condition (blue) and outcomes 0€ (141±42) to –1€ (93±27) in the punishment condition (red), (**E**) Across participants trial-wise reward prediction error (PE) (Reward prediction error, RPE - blue) and punishment PE (PPE - red), ±95% confidence interval.

The online version of this article includes the following figure supplement(s) for figure 1:

**Figure supplement 1.** Single subject anatomical repartition.

**Figure supplement 2.** Single-subject estimation of prediction errors.

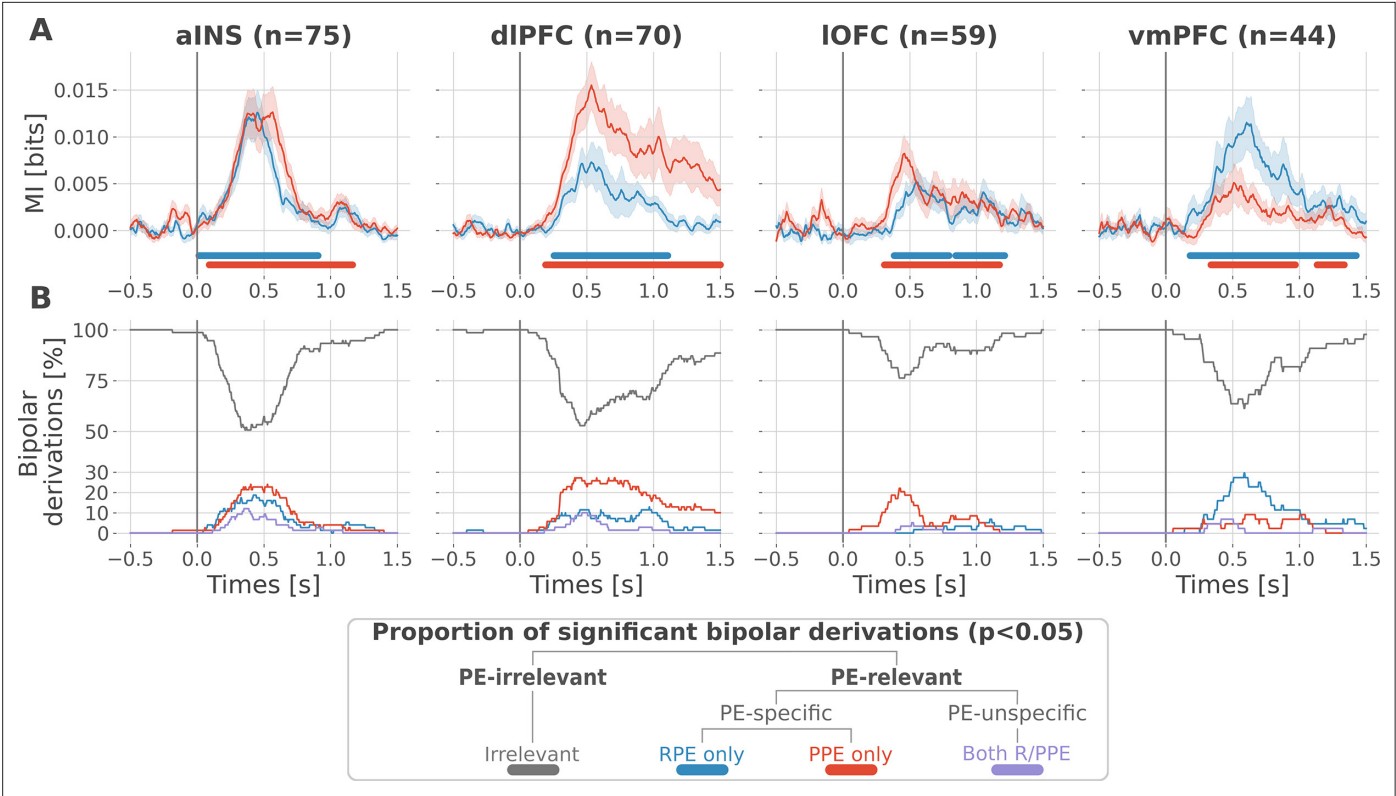

**Figure 2.** Local mixed encoding of reward and punishment prediction error signals. (**A**) Time-courses of mutual information (MI in bits) estimated between the gamma power and the reward (blue) and punishment (red) prediction error (PE) signals. The solid line and the shaded area represent the mean and SEM of the across-contacts MI. Significant clusters of MI at the group level are plotted with horizontal bold lines (p<0.05, cluster-based correction, non-parametric randomization across epochs), (**B**) Instantaneous proportions of task-irrelevant (gray) and task-relevant bipolar derivations presenting a significant relation with either the reward prediction error (RPE) (blue), the punishment prediction error (PPE) (red) or with both RPE and PPE (purple). Data is aligned to the outcome presentation (vertical line at 0 s).

The online version of this article includes the following figure supplement(s) for figure 2:

**Figure supplement 1.** Local encoding of prediction error signals within the gamma band.

**Figure supplement 2.** Inter-subjects reproducibility of local encoding of prediction error (PE) signals.

and punishment PE signaling. Whereas the vmPFC and dlPFC displayed complementary functional preferences for RPE and PPE, respectively, the aINS and the lOFC carried similar amounts of information about both R/PPE (*Figure 2A*).

To better characterize the spatial granularity of PE encoding, we further studied the specificity of individual brain regions by categorizing bipolar derivations as either: (i) RPE-specific; (ii) PPE-specific; (iii) PE-unspecific responding to both R/PPE; (iv) PE-irrelevant (i.e. non-significant ones) (*Figure 2B*). All regions displayed a local mixed encoding of prediction errors with temporal dynamics peaking around 500 ms after outcome presentation. The vmPFC and dlPFC differentially responded to reward and punishment PEs, and contained approximately 30% of RPE- and PPE-specific contacts, respectively. In both regions, the proportion of RPE- and PPE-specific bipolar derivations was elevated for approximately 1 s after outcome presentation. The lOFC also contained a large proportion of PPE-specific bipolar derivations, but displayed more transient dynamics lasting approximately 0.5 s. The aINS had similar proportions of bipolar derivations specific for the RPE and PPE (20%), with temporal dynamics lasting approximately 0.75 s. Importantly, all regions contained approximately 10% of PE-unspecific bipolar derivations that responded to both RPE and PPE, especially in the aINS and dlPFC. The remaining bipolar derivations were categorized as PE-irrelevant. A complementary analysis, conducted to evaluate inter-subject reproducibility, revealed that local encoding in the lOFC and vmPFC was represented in 30 to 50% of the subjects. In contrast, this encoding was found in 50 to 100% of the subjects in the aINS and dlPFC (*Figure 2—figure supplement 2*).

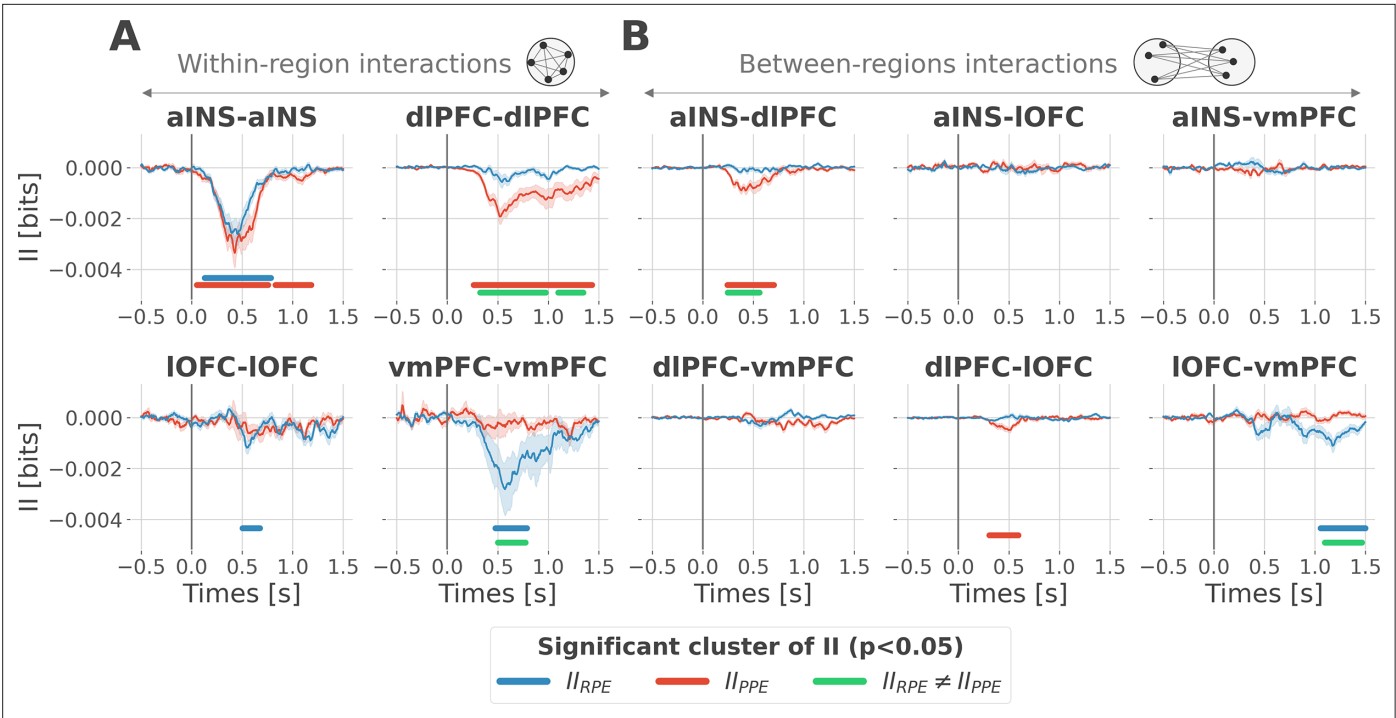

**Figure 3.** Encoding of prediction error (PE) signals occurs with redundancy-dominated subsystems. Dynamic interaction information (II in bits) within-(**A**) and between-regions (**B**) about the RPE (II$_{RPE}$) and PPE (II$_{PPE}$) are plotted in blue and red. Significant clusters of II$_{RPE}$ and II$_{PPE}$ are displayed with horizontal bold blue and red lines (p<0.05, cluster-based correction, non-parametric randomization across epochs). Significant differences between II$_{RPE}$ and II$_{PPE}$ are displayed in green. Shaded areas represent the SEM. The vertical gray line at 0 s represents the outcome presentation.

The online version of this article includes the following figure supplement(s) for figure 3:

**Figure supplement 1.** Inter-subjects reproducibility of redundant interactions about prediction error (PE) signals.

Taken together, our results demonstrate that reward and avoidance learning are not supported by highly selective brain activations, but rather from a mixed or mixed encoding of RPE or PPE signals distributed over the prefrontal and insular cortices. Nevertheless, such distributed encoding seems to involve two complementary systems primarily centered over the vmPC and dlPFC, respectively.

## Encoding of PE signals occurs with redundancy-dominated subsystems

To better understand the observed complex encoding of reward and punishment PEs, we tested the hypothesis that functional dissociations occur with differential and distributed interactions between prefrontal and insular cortices. To address this question, we performed model-based network-level analyses based on the PID framework (*Williams and Beer, 2010*; *Wibral et al., 2017*; *Lizier et al., 2018*). We particularly used the interaction information (*McGill, 1954*; *Ince et al., 2017*) to quantify whether a three-variable interaction (i.e. pairs of brain regions, and the PE variable) is either synergy-and redundancy-dominated (*Williams and Beer, 2010*). Indeed, interaction information (II) can be either positive or negative. A negative value indicates a net redundancy (i.e. a pair of recordings are carrying similar information about the PE), whereas a positive value indicates a net synergistic effect (i.e. a pair of recordings are carrying complementary information about the PE). We computed the time-resolved II across trials between the gamma activity of pairs of iEEG signals and PEs. To differentiate cortico-cortical interactions for reward and punishment learning, we first calculated the II separately for RPEs and PPEs. RPE- and PPE-specific analyses exclusively showed negative modulations of II, therefore, indicating the presence of redundancy-dominated local and long-range interactions (*Figure 3*).

To better characterize the local interactions encoding reward and punishment PEs, we computed the II between pairs of gamma band signals recorded within the aINS, dlPFC, lOFC, and vmPFC. Within-region II analyses showed that significant RPE-specific interactions were exclusively observed in the vmPFC and lOFC, whereas PPE-specific interactions were present only in the dlPFC. In addition,

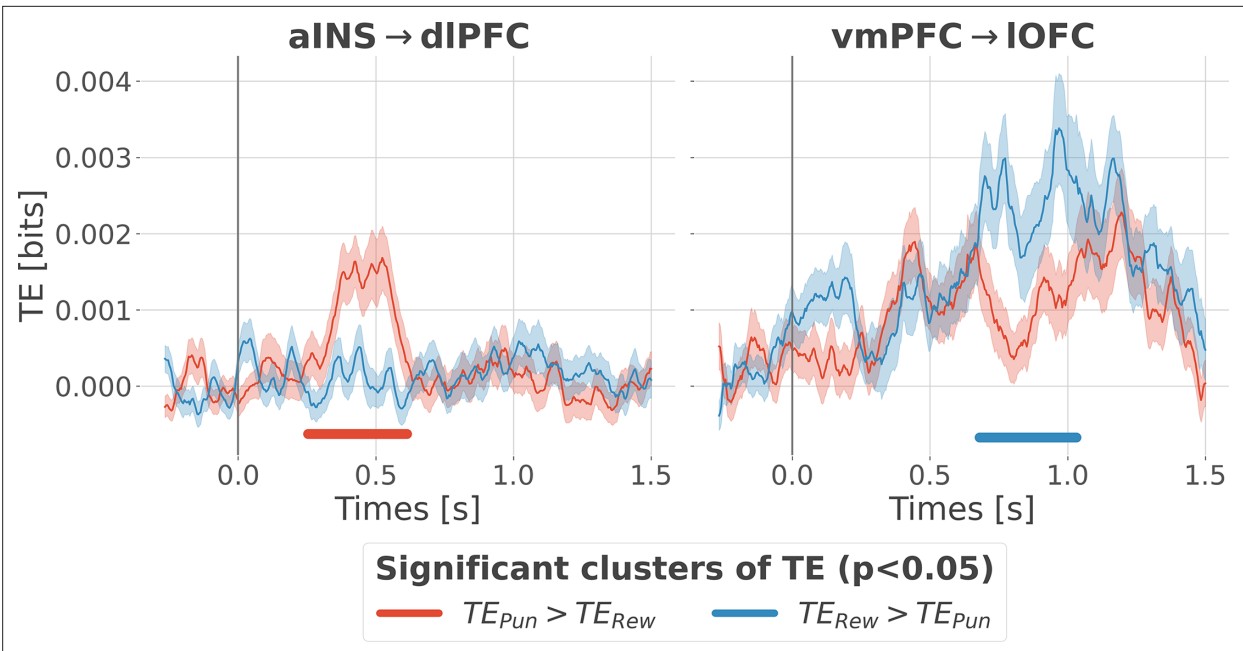

**Figure 4.** Contextual modulation of information transfer. Time courses of transfer entropy (TE, in bits) from the anterior insula (aINS) to the dorsolateral prefrontal cortex (dlPFC) (aINS→dlPFC) and from the vmPFC to the lateral orbitofrontal cortex (lOFC) (vmPFC→lOFC), estimated during the rewarding condition (TE$_{Rew}$ in blue) and punishing condition (TE$_{Pun}$ in red). Significant differences (p<0.05, cluster-based correction, non-parametric randomization across epochs) of TE between conditions are displayed with horizontal bold lines (blue for TE$_{Rew}$ >TE$_{Pun}$ and red for TE$_{Pun}$ >TE$_{Rew}$). Shaded areas represent the SEM. The vertical gray line at 0 s represents the outcome presentation.

The online version of this article includes the following figure supplement(s) for figure 4:

**Figure supplement 1.** Optimal delay interval for maximizing information transfer.

**Figure supplement 2.** Contextual modulation of the information transfer.

the aINS was found to display both RPE- and PPE-specific interactions (*Figure 3A*). A relevant sign of high specificity for either reward or punishment PE signals was the presence of a significant cluster dissociating RPE and PPE in the vmPFC and dlPFC only (green clusters in *Figure 3A*).

To investigate the nature of long-range interactions, we next computed the II for RPE and PPE between signals from different brain regions (*Figure 3B*). Similarly, results exclusively showed redundancy-dominated interactions (i.e. negative modulations). RPE-specific interactions were observed between the lOFC and vmPFC, whereas PPE-specific interactions were observed between the aINS and dlPFC and to a smaller extent between the dlPFC and lOFC, peaking at 500 ms after outcome presentation. A significant difference between RPE and PPE was exclusively observed in the lOFC-vmPFC and aINS-dlPFC interactions, but not between dlPFC and lOFC (green clusters in *Figure 3B*). The analysis of inter-subject reproducibility revealed that both within-area and across-area significant redundant interactions were carried by 30 to 60% of the subjects (*Figure 3—figure supplement 1*). Taken together, we conclude that the encoding of RPE and PPE signals occurs with redundancy-dominated subsystems that differentially engage prefronto-insular regions.

## Contextual directional interactions within redundant subsystems

Previous analyses of II are blind to the direction of information flows. To address this issue, we estimated the transfer entropy (TE) (*Schreiber, 2000*) on the gamma power during the rewarding (TE$_{Rew}$) and punishment conditions (TE$_{Pun}$), between all possible pairs of contacts. As a reminder, the TE is an information-theoretic measure that quantifies the degree of directed statistical dependence or 'information flow' between time series, as defined by the Wiener-Granger principle *Wiener, 1956*; *Granger, 1969*. Delay-specific analyses of TE showed that a maximum delay of information transfer between pairs of signals comprised an interval between 116 and 236 ms (*Figure 4—figure supplement 1*). We thus computed the TE for all pairs of brain regions within this range of delays and detected temporal clusters where the TE significantly differed between conditions (TE$_{Rew}$ >TE$_{Pun}$ or TE$_{Pun}$ >TE$_{Rew}$). Only two

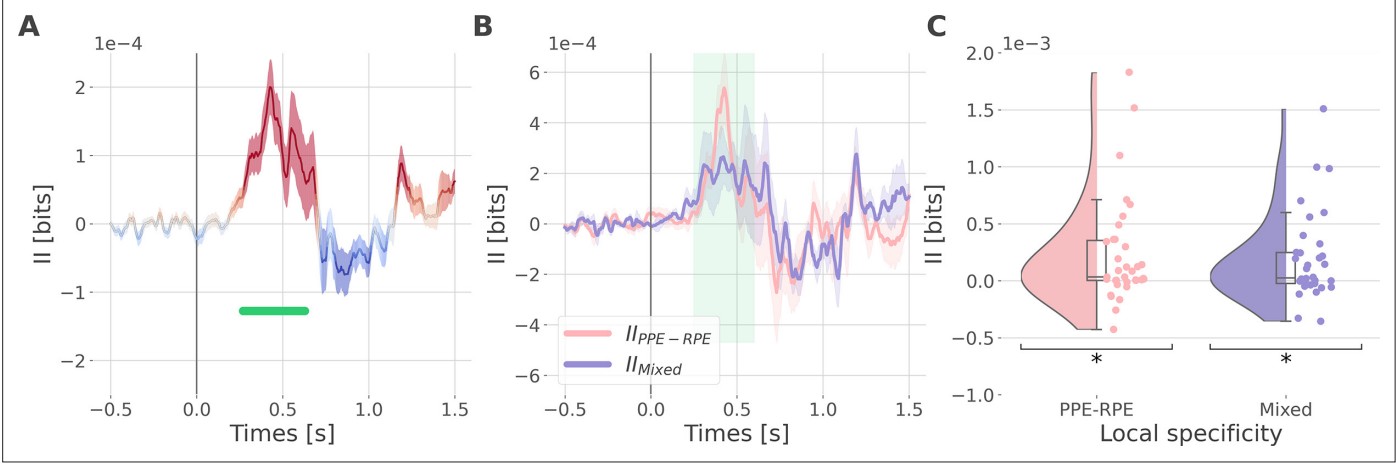

**Figure 5.** Synergistic interactions about the full prediction error (PE) signals between recordings of the dlPFC and vmPFC. (**A**) Dynamic interaction information (II in bits) between the dorsolateral prefrontal cortex (dlPFC) and vmPFC about the full prediction error ($II_{dlPFC-vmPFC}$). Hot and cold colors indicate synergy- and redundancy-dominated II about the full PE. Significant clusters of II are displayed with a horizontal bold green line (p<0.05, cluster-based correction, non-parametric randomization across epochs). Shaded areas represent the SEM. The vertical gray line at 0 s represents the outcome presentation. (**B**) Dynamic $II_{dlPFC-vmPFC}$ binned according to the local specificity PPE-RPE ($II_{PPE-RPE}$ in pink) or mixed ($II_{Mixed}$ in purple) (**C**) Distributions of the mean of the $II_{PPE-RPE}$ and $II_{Mixed}$ for each pair of recordings ($II_{PPE-RPE}$: one-sample t-test against 0; dof = 34; P fdr-corrected=0.015*; T=2.86; CI(95%)=[6.5e-5, 3.9e-4]; $II_{Mixed}$: dof = 33; P fdr-corrected=0.015*; T=2.84; CI(95%)=[5.4e-5, 3.3e-4]).

The online version of this article includes the following figure supplement(s) for figure 5:

**Figure supplement 1.** Cortico-cortical interactions about the full prediction error (PE) signals.

**Figure supplement 2.** Interaction information is binned according to the local specificity.

**Figure supplement 3.** Local specificity does not fully determine the type of interactions.

pairs of brain regions displayed statistically-significant modulations in TE (*Figure 4*). We observed that the TE from the aINS to the dlPFC ($TE_{aINS \to dlPFC}$) peaked at approximately 400 ms after outcome onset and was significantly stronger during the punishment condition compared to the rewarding condition. By contrast, the information flow around ~800 ms from the vmPFC to the lOFC ($TE_{vmPFC \to lOFC}$) was significantly stronger during the rewarding condition. No other brain interactions were found significant (*Figure 4—figure supplement 2*). Overall, these results demonstrate that the two redundancy-dominated RPE- and PPE-specific networks (*Figure 3B*) are characterized by differential directional interactions. The vmPFC and aINS act as drivers in the two systems, whereas the dlPFC and lOFC play the role of receivers, thus suggesting a flow of PE-specific information within the network.

## Integration of PE signals occurs with synergy-dominated interactions between segregated sub-systems

Since learning required participants to concurrently explore rewarding and punishment outcomes, we finally investigated the nature of cortico-cortical interactions encoding both RPE and PPE signals. We estimated the II about the full PEs, i.e., the information carried by co-modulation of gamma power between all pairs of contacts about PE signals (*Figure 5—figure supplement 1*). Encoding of PEs was specifically associated with significantly positive II between the dlPFC and vmPFC ($II_{dlPFC-vmPFC}$ *Figure 5A*). Such between-regions synergy-dominated interaction occurred approximately between 250 and 600 ms after outcome onset.

We then investigated if the synergy between the dlPFC and vmPFC encoding global PEs could be explained by their respective local specificity. Indeed, we previously reported larger proportions of recordings encoding the PPE in the dlPFC and the RPE in the vmPFC (*Figure 2B*). Therefore, it is possible that the positive $II_{dlPFC-vmPFC}$ could be mainly due to complementary roles where the dlPFC brings information about the PPE only and the vmPFC brings information to the RPE only. To test this possibility, we computed the $II_{dlPFC-vmPFC}$ for groups of bipolar derivations with different local specificities. As a reminder, bipolar derivations were previously categorized as RPE or PPE specific if their gamma activity were modulated according to the RPE only, to the PPE only, or to both (*Figure 2B*).

We obtained four categories of II. The first two categories, named $II_{RPE-RPE}$ and $II_{PPE-PPE}$, reflect the II estimated between RPE- and PPE- bipolar derivations from the dlPFC and vmPFC. The third category ($II_{PPE-RPE}$) refers to the II estimated between PPE-specific bipolar recordings from the dlPFC and RPE-specific bipolar recordings from the vmPFC. Finally, the fourth category, named $II_{Mixed}$, includes the remaining possibilities (i.e. RPE-Both, PPE-Both, and Both-Both) (*Figure 5—figure supplement 2*). Interestingly, we found significant synergistic interactions between recordings with mixed specificity i.e., $II_{PPE-RPE}$ and $II_{Mixed}$ between 250 and 600ms after outcome onset (*Figure 5B and C*). Consequently, the $II_{dlPFC-vmPFC}$ is partly explained by the dlPFC and vmPFC carrying PPE- and RPE-specific information ($II_{PPE-RPE}$) together with interactions between non-specific recordings ($II_{Mixed}$). In addition, we simulated data to demonstrate that synergistic interactions can emerge between regions with the same local specificity (*Figure 5—figure supplement 3*). Taken together, the integration of the global PE signals occurred with a synergistic interaction between recordings with mixed specificity from the dlPFC and vmPFC.

## Discussion

Our study revealed the presence of specific functional interactions between prefrontal and insular cortices about reward and punishment prediction error signals. We first provided evidence for a mixed encoding of reward and punishment prediction error signals in each cortical region. We then identified a first subsystem specifically encoding RPEs with emerging redundancy-dominated interactions within and between the vmPFC and lOFC, with a driving role of the vmPFC. A second subsystem specifically encoding PPEs occurred with redundancy-dominated interactions within and between the aINS and dlPFC, with a driving role of the aINS. Switching between the encoding of reward and punishment PEs involved a synergy-dominated interaction between these two systems mediated by interactions between the dlPFC and vmPFC (*Figure 6*).

### Local mixed representations of prediction errors

Amongst the four investigated core-learning regions, the vmPFC was the only region to show a higher group-level preference for RPEs. This supports the notion that the vmPFC is functionally more specialized for the processing outcomes in reward learning, as previously put forward by human fMRI meta-analyses (*Yacubian et al., 2006*; *Diekhof et al., 2012*; *Bartra et al., 2013*; *Garrison et al., 2013*; *Fouragnan et al., 2018*). The dlPFC, instead, showed a stronger selectivity for punishment PE, thus supporting results from fMRI studies showing selective activations for aversive outcomes (*Liu et al., 2011*; *Garrison et al., 2013*; *Fouragnan et al., 2018*). On the contrary, the aINS and lOFC did not show clear selectivity for either reward or punishment PEs. The aINS carried a comparable amount of information about the RPE and PPE, thus suggesting that the insula is part of the surprise-encoding network (*Fouragnan et al., 2018*; *Loued-Khenissi et al., 2020*). Previous study reported a stronger link between the gamma activity of the aINS and the PPE compared to the RPE (*Gueguen et al., 2021*). This discrepancy in the results could be explained by the measures of information we are using here that are able to detect both linear and non-linear relationships between gamma activity and PE signals (*Ince et al., 2017*). The lOFC showed an initial temporal selectivity for PPE followed by a delayed one about the RPE. This is in accordance with fMRI and human intracranial studies which revealed that the lOFC was activated when receiving punishing outcomes, but also contains reward-related information (*O'Doherty et al., 2001*; *Saez et al., 2018*; *Gueguen et al., 2021*).

By taking advantage of the multi-site sampling of iEEG recordings, we quantified the heterogeneity in functional selectivity within each area and showed that the region-specific tendency toward either RPE or PPEs (*Figure 2A*) could be explained by the largest domain-specific proportion of contacts (*Figure 2B*). In other words, if a region showed a larger proportion of contacts being RPE-specific, the amount of information about the RPE at the group-level was also larger. Interestingly, we observed that 5 to 20% of contacts within a given region encoded both the RPE and PPE, thus revealing local mixed representations. Consequently, a strict dichotomous classification of learning-related areas as either reward, and punishment may fail to capture important properties of the individual nodes of the learning circuit, such as the functional heterogeneity in the encoding of PEs. These results suggest that the human prefrontal cortex exhibits a mixed local selectivity for prediction error signals at the mesoscopic scale. This view is in line with recent literature showing that the prefrontal cortex contains

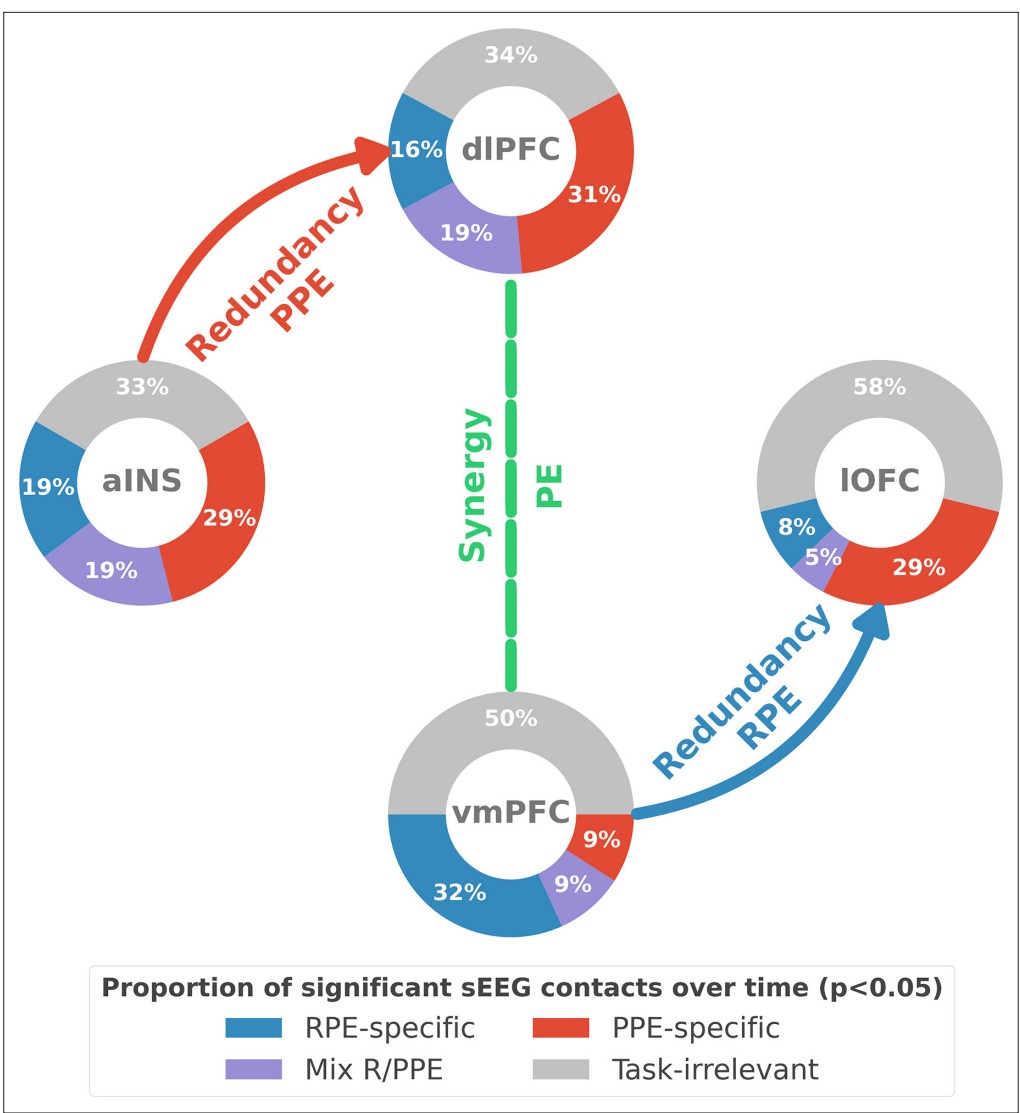

**Figure 6.** Summary of findings. The four nodes represent the investigated regions, namely the anterior insula (aINS), the dorsolateral and ventromedial parts of the prefrontal cortex (dlPFC and vmPFC, and the lateral orbitofrontal cortex lOFC). The outer disc represents the local mixed encoding i.e., the different proportions of contacts over time having a significant relationship between the gamma power and PE signals. In blue, is the proportion of contacts with a significant relation with the PE across rewarding trials (RPE-specific). Conversely, in red for punishment trials (PPE-specific). In purple, the proportion of contacts with a significant relationship with both the reward prediction error (RPE) and punishment prediction error (PPE). In gray, is the remaining proportion of non-significant contacts. Regarding interactions, we found that information transfer between aINS and dlPFC carried redundant information about PPE only and information transfer between vmPFC and lOFC about RPE only. This information transfer occurred with a leading role of the aINS in the punishment context and the vmPFC in the rewarding context. Finally, we found synergistic interactions between the dlPFC and the vmPFC about the full PE, without splitting into rewarding and punishing conditions.

single neurons exhibiting mixed selectivity for multiple task variables (*Meyers et al., 2008*; *Rigotti et al., 2013*; *Stokes et al., 2013*; *Panzeri et al., 2015*; *Parthasarathy et al., 2017*; *Bernardi et al., 2020*). In the learning domain, single-unit studies have reported neurons encoding both rewarding and aversive outcomes in the OFC of the primate (*Morrison and Salzman, 2009*; *Monosov and Hikosaka, 2012*; *Hirokawa et al., 2019*). Mixed selectivity provides computational benefits, such as increasing the number of binary classifications, improving cognitive flexibility, and simplifying readout by downstream neurons (*Fusi et al., 2016*; *Helfrich and Knight, 2019*; *Ohnuki et al., 2021*; *Panzeri et al., 2022*). We suggest that the encoding of cognitive variables such as prediction error signals is

supported by similar principles based on mixed selectivity at the meso- and macroscopic level, and may provide a natural substrate for cognitive flexibility and goal-directed learning (*Rigotti et al., 2013*).

## Redundancy-dominated interactions segregate reward and punishment learning subsystems

We then tested whether the encoding of RPE and PPE signals could be supported by differential cortico-cortical interactions within and between frontal brain regions. To do so, we exploited the interaction information (II) (*McGill, 1954*; *Ince et al., 2017*) to quantify whether the amount of information bound up in a pair of gamma responses and PE signals is dominated by redundant or synergistic interactions (*Williams and Beer, 2010*). The II revealed redundancy-dominated interactions specific for RPE and PPE in the vmPFC and the dlPFC, respectively (*Figure 3A*). The aINS was the only region for which the between-contacts II did not increase the functional selectivity, with large redundant interactions for both RPE and PPE signals. This suggests that within-area redundant interactions can potentially amplify the functional specificity, despite the presence of local mixed selectivity (*Figure 2A*). Such 'winner-take-all' competition could be implemented by mutual inhibition mechanisms, which have been suggested to be essential in reward-guided choice (*Hunt et al., 2012*; *Jocham et al., 2012*; *Strait et al., 2014*; *Hunt and Hayden, 2017*).

Across-areas interaction information revealed two subsystems with redundancy-dominated interactions. A reward subsystem with RPE-specific interactions between the lOFC and vmPFC, and a punishment subsystem with PPE-specific interactions between the aINS and dlPFC (*Figure 3B*). Although a significant modulation selective for RPE was also present in the interaction between dlPFC and lOFC peaking around 500 ms after outcome presentation, a significant difference between the encoding of RPE and PPE was exclusively observed in the lOFC-vmPFC and aINS-dlPFC interactions (green clusters in *Figure 3B*). This result suggests that the observed functionally-distinct learning circuits for RPE and PPEs are associated with differential cortico-cortical interactions, rather than distinct local properties. More generally, our results suggest that redundancy-based network-level interactions are related to the functional specificity observed in neuroimaging and lesion studies (*Pessiglione and Delgado, 2015*; *Palminteri and Pessiglione, 2017*).

We then investigated differential communication patterns and directional relations within the two redundancy-dominated circuits (*Kirst et al., 2016*; *Palmigiano et al., 2017*). We identified significant information routing patterns, and dissociating reward and punishment learning (*Figure 4*). Within the reward subsystem, the vmPFC played a driving role toward the lOFC only during the rewarding condition. Conversely, within the punishment subsystem, the aINS played a driving role toward the dlPFC only during the punishment condition. These results support the notion that redundancy-dominated cognitive networks are associated with the occurrence of information-routing capabilities, where signals are communicated on top of collective reference states (*Battaglia and Brovelli, 2020*).

Here, we quantified directional relationships between regions using the transfer entropy (*Schreiber, 2000*), which is a functional connectivity measure based on the Granger-Wiener causality principle. Tract tracing studies in the macaque have revealed strong interconnections between the lOFC and vmPFC in the macaque (*Carmichael and Price, 1996*; *Ongür and Price, 2000*). In humans, cortico-cortical anatomical connections have mainly been investigated using diffusion magnetic resonance imaging (dMRI). Several studies found strong probabilities of structural connectivity between the anterior insula with the orbitofrontal cortex and the dorsolateral part of the prefrontal cortex (*Cloutman et al., 2012*; *Ghaziri et al., 2017*), and between the lOFC and vmPFC (*Heather Hsu et al., 2020*). In addition, the statistical dependency (e.g. coherence) between the LFP of distant areas could be potentially explained by direct anatomical connections (*Schneider et al., 2021*; *Vinck et al., 2023*). Taken together, the existence of an information transfer might rely on both direct or indirect structural connectivity. However, here we also reported differences in TE between rewarding and punishing trials given the same backbone anatomical connectivity (*Figure 4*). Our results are further supported by a recent study involving drug-resistant epileptic patients with resected insula who showed poorer performance than healthy controls in case of risky loss compared to risky gains (*Von Siebenthal et al., 2017*).

## Encoding the full PE is supported by synergistic interactions between subsystems

Humans can flexibly switch between learning strategies that allow the acquisition of stimulus-action-outcomes associations in changing contexts. We investigated how RPE and PPE subsystems coordinated to allow such behavioral flexibility. To do so, we searched for neural correlates of PEs irrespectively of the context (reward or punishment learning) in between-regions interactions. We found that the encoding of global PE signals was associated with synergy-dominated interactions between the two subsystems, mediated by the interactions between the dlPFC and the vmPFC (*Figure 5*). Importantly, such synergy-dominated interaction reveals that the joint representation of the dlPFC and vmPFC is greater than the sum of their individual contributions to the encoding of global PE signals. Thus, it suggests that successful adaptation in varying contexts requires both the vmPFC and dlPFC for the encoding of global PE signals.

## Role of redundant and synergistic interactions in brain network coordination

At the macroscopic level, few studies investigated the potential role of redundant and synergistic interactions. By combining functional and diffusion MRI, recent work suggested that redundant interactions are predominantly associated with structurally coupled and functionally segregated processing. In contrast, synergistic interactions preferentially support functional integrative processes and complex cognition across higher-order brain networks (*Luppi et al., 2022*). Triadic synergistic interactions between the continuous spike counts recorded within and across areas of the visuo-motor network have been shown to carry behaviorally-relevant information and to display the strongest modulations during the processing of visual information and movement execution (*Varley et al., 2023*). Finally, cortical representations of prediction error signals in the acoustic domain observed tone-related and instantaneous redundant interactions, such as time-lagged synergistic interactions within and across temporal and frontal regions of the auditory system (*Gelens et al., 2023*).

At the microscopic level, the amount of information encoded by a population of neurons can be modulated by pairwise and higher-order interactions, producing varying fractions of redundancy and synergy (*Averbeck et al., 2006*; *Panzeri et al., 2015*; *Panzeri et al., 2022*). Synergistic and redundant pairs of neurons can be identified by estimating the amount of information contained in the joint representation minus the sum of the information carried by individual neurons (*Schneidman et al., 2003*). Redundant coding is intricately linked to correlated activity (*Gutnisky and Dragoi, 2008*) and can spontaneously emerge due to the spatial correlations present in natural scenes by triggering neurons with overlapping receptive fields. Correlations between the trial-by-trial variations of neuronal responses could limit the amount of information encoded by a population (*Bartolo et al., 2020*; *Kafashan et al., 2021*) and facilitate readout by downstream neurons (*Salinas and Sejnowski, 2001*). While redundancy has been at the heart of heated debates and influential theories, such as efficient coding and redundancy compression in sensory areas (*Barlow, 2001*), synergy phenomena have been described to a lesser extent. Recently, a study reported synergistic coding in a V1 cortical column together with structured correlations between synergistic and redundant hubs (*Nigam et al., 2019*). Taken together, we suggest that population codes with balancing proportions of redundancy and synergy offer a good compromise between system robustness and resilience to cell loss and the creation of new information (*Panzeri et al., 2022*). We suggest that redundancy-dominated interactions confer robustness and network-level selectivity for complementary learning processes, which may lead to functional integration processes. On the other hand, synergy-dominated interactions seem to support neural interactions between redundancy-dominated networks, thus supporting functional integrative processes in the brain. In addition, our study suggests that redundant and synergistic interactions occur across multiple spatial scales from local to large-scale.

## Conclusion

Our report of mixed representation of reward and punishment prediction error signals explains the discrepancy in the attribution of a functional specificity to the core learning cortical regions. Instead, we propose that functional specialization for reward and punishment PE signals occurs with redundancy-dominated interactions within the two subsystems formed by the vmPFC-lOFC and aINS-dlPFC, respectively. Within each subsystem, we observed asymmetric and directional interactions

with the vmPFC and aINS playing a driving role in the reward and punishment learning circuits. Finally, switching between reward and punishment learning was supported by synergistic collaboration between subsystems. This supports the idea that higher-order integration between functionally-distinct subsystems are mediated by synergistic interactions. Taken together, our results provide a unifying view reconciling distributed cortical representations with interactions supporting reward and punishment learning. They highlight the relevance of considering learning as a network-level phenomenon by linking distributed and functionally redundant subnetworks through synergistic interactions hence supporting flexible cognition (*Fedorenko and Thompson-Schill, 2014*; *Petersen and Sporns, 2015*; *Bassett and Mattar, 2017*; *Hunt and Hayden, 2017*; *Averbeck and Murray, 2020*; *Averbeck and O'Doherty, 2022*).

## Methods
### Data acquisition and experimental procedure
#### Intracranial EEG recordings
Intracranial electroencephalography (iEEG) recordings were collected from sixteen patients presenting pharmaco-resistant focal epilepsy and undergoing presurgical evaluation (33.5±12.4 years old, 10 females). As the location of the epileptic foci could not be identified through noninvasive methods, neural activity was monitored using intracranial stereotactic electroencephalography. Multi-lead and semi-rigid depth electrodes were stereotactically implanted according to the suspected origin of seizures. The selection of implantation sites was based solely on clinical aspects. iEEG recordings were performed at the clinical neurophysiology epilepsy departments of Grenoble and Lyon Hospitals (France). iEEG electrodes had a diameter of 0.8 mm, 2 mm wide, 1.5 mm apart, and contained 8–18 contact leads (Dixi, Besançon, France). For each patient, 5–17 electrodes were implanted. Recordings were conducted using an audio–video-EEG monitoring system (Micromed, Treviso, Italy), which allowed simultaneous recording of depth iEEG channels sampled at 512 Hz (six patients), or 1024 Hz (12 patients) [0.1–200 Hz bandwidth]. One of the contacts located in the white matter was used as a reference. Anatomical localizations of iEEG contacts were determined based on post-implant computed tomography scans or post-implant MRI scans coregistered with pre-implantation scans (*Lachaux et al., 2003*; *Chouairi et al., 2022*). All patients gave written informed consent and the study received approval from the ethics committee (CPP 09-CHUG-12, study 0907) and from a competent authority (ANSM no: 2009-A00239-48).

#### Limitations
iEEG have been collected from pharmacoresistant epileptic patients who underwent deep electrode probing for preoperative evaluation. However, we interpreted these data as if collected from healthy subjects and assumed that epileptic activity does not affect the neural realization of prediction error. To best address this question, we excluded electrodes contaminated with pathological activity and focused on task-related changes and multi-trial analysis to reduce the impact of incorrect or task-independent neural activations. Therefore, our results may benefit from future replication in healthy controls using non-invasive recordings. Despite the aforementioned limitations, we believe that access to deep intracerebral EEG recordings of human subjects can provide privileged insight into the neural dynamics that regulate human cognition, with outstanding spatial, temporal, and spectral precision. In the long run, this type of data could help bridge the gap between neuroimaging studies and electrophysiological recordings in nonhuman primates.

#### Preprocessing of iEEG data
Bipolar derivations were computed between adjacent electrode contacts to diminish contributions of distant electric sources through volume conduction, reduce artifacts, and increase the spatial specificity of the neural data. Bipolar iEEG signals can approximately be considered as originating from a cortical volume centered within two contacts (*Brovelli et al., 2005*; *Bastin et al., 2016*; *Combrisson et al., 2017*), thus providing a spatial resolution of approximately 1.5–3 mm (*Lachaux et al., 2003*; *Jerbi et al., 2009*; *Chouairi et al., 2022*). Recording sites with artifacts and pathological activity (e.g. epileptic spikes) were removed using visual inspection of all of the traces of each site and each participant.

## Definition of anatomical regions of interest

Anatomical labeling of bipolar derivations was performed using the IntrAnat software (*Deman et al., 2018*). The 3D T1 pre-implantation MRI gray/white matter was segmented and spatially normalized to obtain a series of cortical parcels using MarsAtlas (*Auzias et al., 2016*) and the Destrieux atlas (*Destrieux et al., 2010*). 3D coordinates of electrode contacts were then coregistered on post-implantation images (MRI or CT). Each recording site (i.e. bipolar derivation) was labeled according to its position in a parcellation scheme in the participant's native space. Thus, the analyzed dataset only included electrodes identified to be in the gray matter. Four regions of interest (ROIs) were defined for further analysis: (1) the ventromedial prefrontal cortex (vmPFC) ROI was created by merging six (three per hemisphere) parcels in MarsAlas (labeled PFCvm, OFCv, and OFCvm in MarsAtlas) corresponding to the ventromedial prefrontal cortex and fronto-medial part of the orbitofrontal cortex, respectively; (2) the lateral orbitofrontal cortex (lOFC) ROI included four (two per hemisphere) MarsAtlas parcels (MarsAtlas labels: OFCvl and the OFCv); (3) the dorsolateral prefrontal cortex (dlPFC) ROI was defined as the inferior and superior bilateral dorsal prefrontal cortex (MarsAtlas labels: PFrdli and PFrdls); (4) the anterior insula (aINS) ROI was defined as the bilateral anterior part of the insula (Destrieux atlas labels: Short insular gyri, anterior circular insular sulcus and anterior portion of the superior circular insular sulcus). The total number of bipolar iEEG derivations for the four ROIS was 44, 59, 70, and 75 for the vmPFC, lOFC, dlPFC, and aINS, respectively (*Figure 1A*). As channels with artifacts or epileptic activities were removed here, the number of recordings differs from a previous study (*Gueguen et al., 2021*).

## Behavioral task and set-up

Participants were asked to participate in a probabilistic instrumental learning task adapted from previous studies (*Pessiglione et al., 2006*; *Palminteri et al., 2012*). Participants received written instructions that the goal of the task was to maximize their financial payoff by considering reward-seeking and punishment avoidance as equally important. Instructions were reformulated orally if necessary. Participants started with a short session, with only two pairs of cues presented on 16 trials, followed by 2–3 short sessions of 5 min. At the end of this short training, all participants were familiar with the timing of events, with the response buttons and all reached a threshold of at least 70% of correct choices during both reward and punishment conditions. Participants then performed three to six sessions on a single testing occurrence, with short breaks between sessions. Each session was an independent task, with four new pairs of cues to be learned. Cues were abstract visual stimuli taken from the Agathodaimon alphabet. The two cues of a pair were always presented together on the left and right of a central fixation cross and their relative position was counterbalanced across trials. On each trial, one pair was randomly presented. Each pair of cues was presented 24 times for a total of 96 trials per session. The four pairs of cues were divided into two conditions. A rewarding condition where the two pairs could either lead the participants to win one euro or nothing (+1€ vs. 0€) and a symmetric punishment condition where the participants could either lose one euro or nothing (–1€ vs. 0€). Rewarding and punishing pairs of cues were presented in an intermingled random manner and participants had to learn the four pairs at once. Within each pair, the two cues were associated with the two possible outcomes with reciprocal probabilities (0.75/0.25 and 0.25/0.75). To choose between the left or right cues, participants used their left or right index to press the corresponding button on a joystick (Logitech Dual Action). Since the position on the screen was counterbalanced, response (left versus right) and value (good vs. bad cue) were orthogonal. The chosen cue was colored in red for 250 ms and then the outcome was displayed on the screen after 1000 ms. To win money, participants had to learn by trial and error which cue-outcome association was the most rewarding in the rewarding condition and the least penalizing in the punishment condition. Visual stimuli were delivered on a 19-inch TFT monitor with a refresh rate of 60 Hz, controlled by a PC with Presentation 16.5 (Neurobe-havioral Systems, Albany, CA).

## Computational model of learning

To model choice behavior and estimate prediction error signals, we used a standard Q-learning model (*Watkins and Dayan, 1992*) from reinforcement learning theory (*Sutton and Barto, 2018*). For a pair of cues $A$ and $B$, the model estimates the expected value of choosing $A$ ($Qa$) or $B$ ($Qb$), given previous choices and received outcomes. Q-values were initiated to 0, corresponding to the average of all

**Table 1.** Results of the one-sample t-test performed against 0.

|  | T-value | p-value | p-value (FDR corrected) | dof | CI 95% |
|---|---|---|---|---|---|
| $II_{PPE-RPE}$ | 2859 | 0.007** | 0.015* | 34 | [6.5e-05, 3.9e-04] |
| $II_{Mixed}$ | 2841 | 0.008** | 0.015* | 33 | [5.4e-05, 3.3e-04] |
| $II_{PPE-PPE}$ | 1,25 | 0.2667 | 0.3556 | 5 | [–7.1e-05, 2.1e-04] |
| $II_{RPE-RPE}$ | 0733 | 0.4912 | 0.4912 | 6 | [–3.1e-05, 5.8e-05] |

possible outcome values. After each trial *t*, the expected value of choosing a stimulus (e.g. *A*) was updated according to the following update rule:

$$Q_{a_{t+1}} = Q_{a_t} + \alpha\delta_t \tag{1}$$

with α the learning rate weighting the importance given to new experiences and δ, the outcome prediction error signals at a trial *t* defined as the difference between the obtained and expected outcomes:

$$\delta_t = R_t - Qa_t \tag{2}$$

with $R_t$ the reinforcement value among –1€, 0€, and 1€. The probability of choosing a cue was then estimated by transforming the expected values associated with each cue using a softmax rule with a Gibbs distribution. An additional θ parameter was added in the softmax function to the expected value of the chosen option on the previous trial of the same cue to account for the tendency to repeat the choice made on the previous trial. For example, if a participant chose option *A* on trial *t*, the probability of choosing *A* at trial *t+1* was obtained using:

$$Pa_{t+1} = \frac{e^{Qa_t+\theta/\beta}}{e^{Qa_t+\theta/\beta}+e^{Qb_t/\beta}} \tag{3}$$

with β the choice temperature for controlling the ratio between exploration and exploitation. The three free parameters α, β, and θ were fitted per participant and optimized by minimizing the negative log-likelihood of choice using the MATLAB *fmincon* function, initialized at multiple starting points of the parameter space (*Palminteri et al., 2015*). Estimates of the free parameters, the goodness of fit and the comparison between modeled and observed data can be seen in *Table 1* and Figure 1 in *Gueguen et al., 2021*.

## iEEG data analysis
### Estimate of single-trial gamma-band activity

Here, we focused solely on broadband gamma for three main reasons. First, it has been shown that the gamma band activity correlates with both spiking activity and the BOLD fMRI signals (*Mukamel et al., 2005*; *Niessing et al., 2005*; *Lachaux et al., 2007*; *Nir et al., 2007*), and it is commonly used in MEG and iEEG studies to map task-related brain regions (*Brovelli et al., 2005*; *Crone et al., 2006*; *Vidal et al., 2006*; *Ball et al., 2008*; *Jerbi et al., 2009*; *Lachaux et al., 2012*; *Cheyne and Ferrari, 2013*). Therefore, focusing on the gamma band facilitates linking our results with the fMRI and spiking literature on probabilistic learning. Second, single-trial and time-resolved high-gamma activity can be exploited for the analysis of cortico-cortical interactions in humans using MEG and iEEG techniques (*Brovelli et al., 2015*; *Brovelli et al., 2017*; *Combrisson et al., 2022a*). Finally, while previous analyses of the current dataset (*Gueguen et al., 2021*) reported an encoding of PE signals at different frequency bands, the power in lower frequency bands were shown to carry redundant information compared to the gamma band power. In the current study, we thus estimated the power in the gamma band using a multitaper time-frequency transform based on Slepian tapers (*Percival and Walden, 1993*; *Mitra and Pesaran, 1999*). To extract gamma-band activity from 50 to 100 Hz, the iEEG time series were multiplied by 9 orthogonal tapers (15 cycles for a duration of 200ms and with a time-bandwidth for frequency smoothing of 10 Hz), centered at 75 Hz and Fourier-transformed. To limit false negative proportions due to multiple testings, we down-sampled the gamma power

to 256 Hz. Finally, we smoothed the gamma power using a 10-point Savitzky-Golay filter. We used MNE-Python (*Gramfort et al., 2013*) to inspect the time series, reject contacts contaminated with pathological activity, and estimate the power spectrum density (mne.time_frequency.psd_multitaper) and the gamma power (mne.time_frequency.tfr_multitaper).

## Local correlates of PE signals

To quantify the local encoding of prediction error (PE) signals in the four ROIs, we used information-theoretic metrics. To this end, we computed the time-resolved mutual information (MI) between the single-trial gamma-band responses and the outcome-related PE signals. As a reminder, mutual information is defined as:

$$I(X, Y) = H(X) - H(X|Y) \tag{4}$$

In this equation, the variables X and Y represent the across-trials gamma-band power and the PE variables, respectively. H(X) is the entropy of X, and H(X|Y) is the conditional entropy of X given Y. In the current study, we used a semi-parametric binning-free technique to calculate MI, called Gaussian-Copula Mutual Information (GCMI) (*Ince et al., 2017*). The GCMI is a robust rank-based approach that allows the detection of any type of monotonic relation between variables and it has been successfully applied to brain signals analysis (*Colenbier et al., 2020*; *Michelmann et al., 2021*; *Ten Oever et al., 2021*). Mathematically, the GCMI is a lower-bound estimation of the true MI and it does not depend on the marginal distributions of the variables, but only on the copula function that encapsulates their dependence. The rank-based copula-normalization preserves the relationship between variables as long as this relation is strictly increasing or decreasing. As a consequence, the GCMI can only detect monotonic relationships. Nevertheless, the GCMI is of practical importance for brain signal analysis for several reasons. It allows to estimate the MI on a limited number of samples and it contains a parametric bias correction to compensate for the bias due to the estimation on smaller datasets. It allows to compute the MI on uni- and multivariate variables that can either be continuous or discrete see *Table 1* in *Ince et al., 2017*. Finally, it is computationally efficient, which is a desired property when dealing with a large number of iEEG contacts recording at a high sampling rate. Here, the GCMI was computed across trials and it was used to estimate the instantaneous amount of information shared between the gamma power of iEEG contacts and RPE ($MI_{RPE} = I(\gamma; RPE)$) and PPE signals ($M_{PPE} = I(\gamma; PPE)$).

## Network-level interactions and PE signals

The goal of network-level analyses was to characterize the nature of cortico-cortical interactions encoding reward and punishment PE signals. In particular, we aimed to quantify: (1) the nature of the interdependence between pairs of brain ROIs in the encoding of PE signals; (2) the information flow between ROIs encoding PE signals. These two questions were addressed using Interaction Information and Transfer Entropy analyses, respectively.

## Interaction Information analysis

In classical information theory, interaction information (II) provides a generalization of mutual information for more than two variables (*McGill, 1954*; *Ince et al., 2017*). For the three-variables case, the II can be defined as the difference between the total, or joint, mutual information between ROIs ($R_1$ and $R_2$) and the third behavioral variable (S), minus the two individual mutual information between each ROI and the behavioral variable. For a three variables multivariate system composed of two sources $R_1$, $R_2$, and a target S, the II is defined as:

$$\begin{aligned} II(R_1; R_2; S) \quad &= I(S; R_1 \mid R_2) - I(R_1; S) \\ &= I(R_1, R_2; S) - I(R_1; S) - (R_2; S) \end{aligned} \tag{5}$$

Unlike mutual information, the interaction information can be either positive or negative. A negative value of interaction information indicates a net redundant effect between variables, whereas positive values indicate a net synergistic effect (*Williams and Beer, 2010*). Here, we used the II to investigate the amount of information and the nature of the interactions between the gamma power of pairs of contacts ($\gamma_1$, $\gamma_2$) about the RPE ($II_{RPE} = II(\gamma_1, \gamma_2; RPE)$) and PPE signals ($II_{PPE} = II(\gamma_1, \gamma_2; PPE)$).

The II was computed by estimating the MI quantities of equation (5) using the GCMI between contacts within the same brain region or across different regions.

## Transfer entropy analysis

To quantify the degree of communication between neural signals, the most successful model-free methods rely on the Wiener-Granger principle (*Wiener, 1956*; *Granger, 1969*). This principle identifies information flow between time series when future values of a given signal can be predicted from the past values of another, above and beyond what can be achieved from its autocorrelation. One of the most general information theoretic measures based on the Wiener-Granger principle is Transfer Entropy (TE) (*Schreiber, 2000*). The TE can be formulated in terms of conditional mutual information (*Schreiber, 2000*; *Kaiser and Schreiber, 2002*):

$$TE(X \rightarrow Y) = I(X_{Past}; Y_t \mid Y_{past}) \tag{6}$$

Here, we computed the TE on the gamma activity time courses of pairs of iEEG contacts. We used the GCMI to estimate conditional mutual information. For an interval $[d_1, d_2]$ of $n_{delays}$, the final TE estimation was defined as the mean over the TE estimated at each delay:

$$TE(X \rightarrow Y)_{[d_1, d_2]} = \frac{1}{n_{delays}} \cdot \sum_{d=d_1}^{d_2} I(X_d; Y_t \mid Y_d) \tag{7}$$

## Statistical analysis

We used a group-level approach based on non-parametric permutations, encompassing non-negative measures of information (*Combrisson et al., 2022a*). The same framework was used at the local level (i.e. the information carried by a single contact) or at the network level (i.e. the information carried by pairs of contacts for the II and TE). To take into account the inherent variability existing at the local and network levels, we used a random-effect model. To generate the distribution of permutations at the local level, we shuffled the PE variable across trials 1000 times and computed the MI between the gamma power and the shuffled version of the PE. The shuffling led to a distribution of MI reachable by chance, for each contact and at each time point (*Combrisson and Jerbi, 2015*). To form the group-level effect, we computed a one-sample t-test against the permutation mean across the MI computed on individual contacts taken from the same brain region, at each time point. The same procedure was used on the permutation distribution to form the group-level effect reachable by chance. We used cluster-based statistics to correct for multiple comparisons (*Maris and Oostenveld, 2007*). The cluster-forming threshold was defined as the 95th percentile of the distribution of t-values obtained from the permutations. We used this threshold to form the temporal clusters within each brain region. We obtained cluster masses on both the true t-values and the t-values computed on the permutations. To correct for multiple comparisons, we built a distribution made of the largest 1000 cluster masses estimated on the permuted data. The final corrected p-values were inferred as the proportion of permutations exceeding the t-values. To generate the distributions of II and TE reachable by chance, we respectively shuffled the PE variable across trials for the II and the gamma power across trials of the source for the TE (*Vicente et al., 2011*). The rest of the significance testing procedure at the network level is similar to the local level, except that it is not applied within brain regions but within pairs of brain regions.

## Software

Information-theoretic metrics and group-level statistics, are implemented in a homemade Python software called *Frites* (*Combrisson et al., 2022b*). The interaction information can be computed using the frites.conn.conn_ii function and the transfer entropy using the frites.conn.conn_te function.

# Acknowledgements

We would like to express our gratitude to Benjamin Morillon, Manuel R Mercier, and Stefano Palminteri for their valuable comments on an earlier draft of this manuscript and for their feedback on our responses to reviewer comments.

# Additional information

## Funding

| Funder | Grant reference number | Author |
|---|---|---|
| Agence Nationale de la Recherche | ANR-18-CE28-0016 | Etienne Combrisson<br>Ruggero Basanisi<br>Julien Bastin<br>Andrea Brovelli |
| Agence Nationale de la Recherche | ANR-17-CE37-0018 | Maelle CM Gueguen<br>Sylvain Rheims<br>Philippe Kahane<br>Julien Bastin |
| Agence Nationale de la Recherche | ANR- 13-TECS-0013 | Maelle CM Gueguen<br>Sylvain Rheims<br>Philippe Kahane<br>Julien Bastin |
| HORIZON EUROPE Framework Programme | 604102 | Julien Bastin |
| HORIZON EUROPE Framework Programme | 945539 | Etienne Combrisson<br>Andrea Brovelli |

The funders had no role in study design, data collection and interpretation, or the decision to submit the work for publication.

## Author contributions

Etienne Combrisson, Conceptualization, Data curation, Software, Formal analysis, Investigation, Visualization, Methodology, Writing – original draft, Writing – review and editing; Ruggero Basanisi, Software, Methodology; Maelle CM Gueguen, Conceptualization, Data curation; Sylvain Rheims, Philippe Kahane, Data curation; Julien Bastin, Data curation, Supervision, Funding acquisition, Writing – review and editing; Andrea Brovelli, Conceptualization, Resources, Software, Supervision, Funding acquisition, Investigation, Methodology, Writing – original draft, Project administration, Writing – review and editing

## Author ORCIDs

Etienne Combrisson (iD) http://orcid.org/0000-0002-7362-3247
Julien Bastin (iD) https://orcid.org/0000-0002-0533-7564
Andrea Brovelli (iD) https://orcid.org/0000-0002-5342-1330

## Ethics

All patients gave written informed consent and the study received approval from the ethics committee (CPP 09-CHUG-12, study 0907) and from a competent authority (ANSM no: 2009-A00239-48).

Reviewer #1 (Public Review): https://doi.org/10.7554/eLife.92938.3.sa1
Reviewer #2 (Public Review): https://doi.org/10.7554/eLife.92938.3.sa2
Reviewer #3 (Public Review): https://doi.org/10.7554/eLife.92938.3.sa3
Author response https://doi.org/10.7554/eLife.92938.3.sa4

# Additional files

## Supplementary files

• MDAR checklist

## Data availability

The Python scripts and notebooks to reproduce the results presented here are hosted on GitHub, copy archived at *Combrisson, 2024*. The preprocessed data used here can be downloaded from Dryad.

The following dataset was generated:

| Author(s) | Year | Dataset title | Dataset URL | Database and Identifier |
|---|---|---|---|---|
| Combrisson E, Basanisi R, Gueguen MCM, Rheims S, Kahane P, Bastin J, Brovelli A | 2024 | Neural interactions in the human frontal cortex dissociate reward and punishment learning | https://datadryad.org/stash/share/hc81iiZcl28Qfd0OjgeCFPIsAL3xwdJtLjU7dFtILZo | Dryad Digital Repository, 10.5061/dryad.jdfn2z3k4 |

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
