## [Editor Report · eLife assessment]

This is an **important** information-theoretic re-analysis of human intracranial recordings during reward and punishment learning. It provides **convincing** evidence that reward and punishment learning is represented in overlapping regions of the brain while relying on specific inter-regional interactions. This preprint will be interesting to researchers in systems and cognitive neuroscience.

---

## [Referee Report · Reviewer #1 (Public Review)]

Summary:

The work by Combrisson and colleagues investigates the degree to which reward and punishment learning signals overlap in the human brain using intracranial EEG recordings. The authors used information theory approaches to show that local field potential signals in the anterior insula and the three sub regions of the prefrontal cortex encode both reward and punishment prediction errors, albeit to different degrees. Specifically, the authors found that all four regions have electrodes that can selectively encode either the reward or the punishment prediction errors. Additionally, the authors analyzed the neural dynamics across pairs of brain regions and found that the anterior insula to dorsolateral prefrontal cortex neural interactions were specific for punishment prediction errors whereas the ventromedial prefrontal cortex to lateral orbitofrontal cortex interactions were specific to reward prediction errors. This work contributes to the ongoing efforts in both systems neuroscience and learning theory by demonstrating how two differing behavioral signals can be differentiated to a greater extent by analyzing neural interactions between regions as opposed to studying neural signals within one region.

Strengths:

The experimental paradigm incorporates both a reward and punishment component that enables investigating both types of learning in the same group of subjects allowing direct comparisons.

The use of intracranial EEG signals provides much needed insight into the timing of when reward and punishment prediction errors signals emerge in the studied brain regions.

Information theory methods provide important insight into the interregional dynamics associated with reward and punishment learning and allows the authors to assess that reward versus punishment learning can be better dissociated based on interregional dynamics over local activity alone.

Weaknesses:

The analysis presented in the manuscript focuses on gamma band activity. Studying slow oscillations could provide additional insights into the interregional dynamics.

---

## [Referee Report · Reviewer #2 (Public Review)]

Reward and punishment learning have long been seen as emerging from separate networks of frontal and subcortical areas, often studied separately. Nevertheless, both systems are complimentary and distributed representations of reward and punishments have been repeatedly observed within multiple areas. This raised the unsolved question of the possible mechanisms by which both systems might interact, which this manuscript went after. The authors skillfully leveraged intracranial recordings in epileptic patients performing a probabilistic learning task combined with model-based information theoretical analyses of gamma activities to reveal that information about reward and punishment was not only distributed across multiple prefrontal and insular regions, but that each system showed specific redundant interactions. The reward subsystem was characterized by redundant interactions between orbitofrontal and ventromedial prefrontal cortex, while the punishment subsystem relied on insular and dorsolateral redundant interactions. Finally, the authors revealed a way by which the two systems might interact, through synergistic interaction between ventromedial and dorsolateral prefrontal cortex.

Here, the authors performed an excellent reanalysis of a unique dataset using innovative approaches, pushing our understanding on the interaction at play between prefrontal and insular cortex regions during learning. Importantly, the description of the methods and results is truly made accessible, making it an excellent resource to the community. The authors also carefully report individual subjects' data, which brings confidence in the reproducibility of their observations.

This manuscript goes beyond what is classically performed using intracranial EEG dataset, by not only reporting where a given information, like reward and punishment prediction errors, is represented but also by characterizing the functional interactions that might underlie such representations. The authors highlight the distributed nature of frontal cortex representations and proposed new ways by which the information specifically flows between nodes. This work is well placed to unify our understanding of the complementarity and specificity of the reward and punishment learning systems.

---

## [Referee Report · Reviewer #3 (Public Review)]

Summary:

The authors investigated that learning processes relied on distinct reward or punishment outcomes in probabilistic instrumental learning tasks were involved in functional interactions of two different cortico-cortical gamma-band modulations, suggesting that learning signals like reward or punishment prediction errors can be processed by two dominated interactions, such as areas lOFC-vmPFC and areas aINS-dlPFC, and later on integrated together in support of switching conditions between reward and punishment learning. By performing the well-known analyses of mutual information, interaction information, and transfer entropy, the conclusion was accomplished by identifying directional task information flow between redundancy-dominated and synergy-dominated interactions. Also, this integral concept provided a unifying view to explain how functional distributed reward and/or punishment information were segregated and integrated across cortical areas.

Strengths:

The dataset used in this manuscript may come from previously published works (Gueguen et al., 2021) or from the same grant project due to the methods. Previous works have shown strong evidence about why gamma-band activities and those 4 areas are important. For further analyses, the current manuscript moved the ideas forward to examine how reward/punishment information transfer between recorded areas corresponding to the task conditions. The standard measurements such mutual information, interaction information, and transfer entropy showed time-series activities in the millisecond level and allowed us to learn the directional information flow during a certain window. In addition, the diagram in Figure 6 summarized the results and proposed an integral concept with functional heterogeneities in cortical areas. These findings in this manuscript will support the ideas from human fMRI studies and add a new insight to electrophysiological studies with the non-human primates.

Comments on revised version:

Thank you authors for all efforts to answer questions from previous comments. I appreciated that authors clarified the terminology and added a paragraph to discuss the current limitations of functional connectivity and anatomical connections. This provided clear and fair explanations to readers who are not familiar with methods in systems neuroscience.

---

## [Author Response]

The following is the authors’ response to the original reviews.

**Public Reviews:**

**Reviewer #1:**
Summary:The work by Combrisson and colleagues investigates the degree to which reward and punishment learning signals overlap in the human brain using intracranial EEG recordings. The authors used information theory approaches to show that local ﬁeld potential signals in the anterior insula and the three sub regions of the prefrontal cortex encode both reward and punishment prediction errors, albeit to different degrees. Speciﬁcally, the authors found that all four regions have electrodes that can selectively encode either the reward or the punishment prediction errors. Additionally, the authors analyzed the neural dynamics across pairs of brain regions and found that the anterior insula to dorsolateral prefrontal cortex neural interactions were speciﬁc for punishment prediction errors whereas the ventromedial prefrontal cortex to lateral orbitofrontal cortex interactions were speciﬁc to reward prediction errors. This work contributes to the ongoing efforts in both systems neuroscience and learning theory by demonstrating how two differing behavioral signals can be differentiated to a greater extent by analyzing neural interactions between regions as opposed to studying neural signals within one region.Strengths:The experimental paradigm incorporates both a reward and punishment component that enables investigating both types of learning in the same group of subjects allowing direct comparisons.The use of intracranial EEG signals provides much needed insight into the timing of when reward and punishment prediction errors signals emerge in the studied brain regions.Information theory methods provide important insight into the interregional dynamics associated with reward and punishment learning and allows the authors to assess that reward versus punishment learning can be better dissociated based on interregional dynamics over local activity alone.

We thank the reviewer for this accurate summary. Please ﬁnd below our answers to the weaknesses raised by the reviewer.

Weaknesses:The analysis presented in the manuscript focuses solely on gamma band activity. The presence and potential relevance of other frequency bands is not discussed. It is possible that slow oscillations, which are thought to be important for coordinating neural activity across brain regions could provide additional insight.

We thank the reviewer for pointing us to this missing discussion in the ﬁrst version of the manuscript. We now made this point clearer in the Methods sections entitled “iEEG data analysis” and “Estimate of single-trial gamma-band activity”:

“Here, we focused solely on broadband gamma for three main reasons. First, it has been shown that the gamma band activity correlates with both spiking activity and the BOLD fMRI signals (Lachaux et al., 2007; Mukamel et al., 2004; Niessing et al., 2005; Nir et al., 2007), and it is commonly used in MEG and iEEG studies to map task-related brain regions (Brovelli et al., 2005; Crone et al., 2006; Vidal et al., 2006; Ball et al., 2008; Jerbi et al., 2009; Darvas et al., 2010; Lachaux et al., 2012; Cheyne and Ferrari, 2013; Ko et al., 2013). Therefore, focusing on the gamma band facilitates linking our results with the fMRI and spiking literatures on probabilistic learning. Second, single-trial and time-resolved high-gamma activity can be exploited for the analysis of cortico-cortical interactions in humans using MEG and iEEG techniques (Brovelli et al., 2015; 2017; Combrisson et al., 2022). Finally, while previous analyses of the current dataset (Gueguen et al., 2021) reported an encoding of PE signals at different frequency bands, the power in lower frequency bands were shown to carry redundant information compared to the gamma band power.”

The data is averaged across all electrodes which could introduce biases if some subjects had many more electrodes than others. Controlling for this variation in electrode number across subjects would ensure that the results are not driven by a small subset of subjects with more electrodes.

We thank the reviewer for raising this important issue. We would like to point out that the gamma activity was not averaged across bipolar recordings within an area, nor measures of connectivity. Instead, we used a statistical approach proposed in a previous paper that combines non-parametric permutations with measures of information (Combrisson et al.,
2022). As we explain in the “Statistical analysis” section, mutual information (MI) is estimated between PE signals and single-trial modulations in gamma activity separately for each contact (or for each pair of contacts). Then, a one-sample t-test is computed across all of the recordings of all subjects to form the effect size at the group-level. We will address the point of the electrode number in our answer below.

The potential variation in reward versus punishment learning across subjects is not included in the manuscript. While the time course of reward versus punishment prediction errors is symmetrical at the group level, it is possible that some subjects show faster learning for one versus the other type which can bias the group average. Subject level behavioral data along with subject level electrode numbers would provide more convincing evidence that the observed effects are not arising from these potential confounds.

We thank the reviewer for the two points raised. We performed additional analyses at the single-participant level to address the issues raised by the reviewer. We should note, however, that these results are descriptive and cannot be generalized to account for population-level effects. As suggested by the reviewer, we prepared two new ﬁgures. The ﬁrst supplementary ﬁgure summarizes the number of participants that had iEEG contacts per brain region and pair of brain regions (Fig. S1A in the Appendix). It can be seen that the number of participants sampled in different brain regions is relatively constant (left panel) and the number of participants with pairs of contacts across brain regions is relatively homogeneous, ranging from 7 to 11 (right panel). Fig. S1B shows the number of bipolar derivations per subject and per brain region.

**Author response image 1. sa4fig1:** Single subject anatomical repartition. (A) Number of unique subject per brain region and per pair of brain regions. (B) Number of bipolar derivations per subject and per brain region.

The second supplementary ﬁgure describes the estimated prediction error for rewarding and punishing trials for each subject (Fig. S2). The single-subject error bars represent the 95th percentile conﬁdence interval estimated using a bootstrap approach across the different pairs of stimuli presented during the three to six sessions. As the reviewer anticipated, there are indeed variations across subjects, but we observe that RPE and PPE are relatively symmetrical, even at the subject level, and tend toward zero around trial number 10. These results therefore corroborate the patterns observed at the group-level.

**Author response image 2. sa4fig2:** Single-subject estimation of predictions errors. Single-subject trial-wise reward PE (RPE - blue) and punishment PE (PPE - red), ± 95% confidence interval.

Finally, to assess the variability of local encoding of prediction errors across participants, we quantiﬁed the proportion of subjects having at least one signiﬁcant bipolar derivation encoding either the RPE or PPE (Fig. S4). As expected, we found various proportions of unique subjects with signiﬁcant R/PPE encoding per region. The lowest proportion was achieved in the ventromedial prefrontal cortex (vmPFC) and lateral orbitofrontal cortex (lOFC) for encoding PPE and RPE, respectively, with approximately 30% of the subjects having the effect. Conversely, we found highly reproducible encodings in the anterior insula (aINS) and dorsolateral prefrontal cortex (dlPFC) with a maximum of 100% of the 9 subjects having at least one bipolar derivation encoding PPE in the dlPFC.

**Author response image 3. sa4fig3:** 

Taken together, we acknowledge a certain variability per region and per condition. Nevertheless, the results presented in the supplementary ﬁgures suggest that the main results do not arise from a minority of subjects.

We would like to point out that in order to assess across-subject variability, a much larger number of participants would have been needed, given the low signal-to-noise ratios observed at the single-participant level. We thus prefer to add these results as supplementary material in the Appendix, rather than in the main text.

It is unclear if the ﬁndings in Figures 3 and 4 truly reﬂect the differential interregional dynamics in reward versus punishment learning or if these results arise as a statistical byproduct of the reward vs punishment bias observed within each region. For instance, the authors show that information transfer from anterior insula to dorsolateral prefrontal cortex is speciﬁc to punishment prediction error. However, both anterior insula and dorsolateral prefrontal cortex have higher prevalence of punishment prediction error selective electrodes to begin with. Therefore the ﬁndings in Fig 3 may simply be reﬂecting the prevalence of punishment speciﬁcity in these two regions above and beyond a punishment speciﬁc neural interaction between the two regions. Either mathematical or analytical evidence that assesses if the interaction effect is simply reﬂecting the local dynamics would be important to make this result convincing.

This is an important point that we partly addressed in the manuscript. More precisely, we investigated whether the synergistic effects observed between the dlPFC and vmPFC encoding global PEs (Fig. 5) could be explained by their respective local speciﬁcity. Indeed, since we reported larger proportions of recordings encoding the PPE in the dlPFC and the RPE in the vmPFC (Fig. 2B), we checked whether the synergy between dlPFC and vmPFC could be mainly due to complementary roles where the dlPFC brings information about the PPE only and the vmPFC brings information to the RPE only. To address this point, we selected PPE-speciﬁc bipolar derivations from the dlPFC and RPE-speciﬁc from the vmPFC and, as the reviewer predicted, we found synergistic II between the two regions probably mainly because of their respective speciﬁcity. In addition, we included the II estimated between non-selective bipolar derivations (i.e. recordings with signiﬁcant encoding for both RPE and PPE) and we observed synergistic interactions (Fig. 5C and Fig. S9). Taken together, the local speciﬁcity certainly plays a role, but this is not the only factor in deﬁning the type of interactions.

Concerning the interaction information results (II, Fig. 3), several lines of evidence suggest that local speciﬁcity cannot account alone for the II effects. For example, the local speciﬁcity for PPE is observed across all four areas (Fig. 2A) and the percentage of bipolar derivations displaying an effect is large (equal or above 10%) for three brain regions (aINS, dlPLF and lOFC). If the local speciﬁcity were the main driving cause, we would have observed signiﬁcant redundancy between all pairs of brain regions. On the other hand, the interaction between the aINS and lOFC displayed no signiﬁcant redundant effect (Fig. 3B). Another example is the result observed in lOFC: approximately 30% of bipolar derivations display a selectivity for PPE (Fig. 2B, third panel from the left), but do not show clear signs of redundant encoding at the level of within-area interactions (Fig. 3A, bottom-left panel). Similarly, the local encoding for RPE is observed across all four brain regions (Fig. 2A) and the percentage of bipolar derivations displaying an effect is large (equal or above 10%) for three brain regions (aINS, dlPLF and vmPFC). Nevertheless, signiﬁcant between-regions interactions have been observed only between the lOFC and vmPFC (Fig. 3B bottom right panel).

To further support the reasoning, we performed a simulation to show that it is possible to observe synergistic interactions between two regions with the same speciﬁcity. As an example, we may consider one region locally encoding early trials of RPE and a second region encoding the late trials of the RPE. Combining the two with the II would lead to synergistic interactions, because each one of them carries information that is not carried by the other. To illustrate this point, we simulated the data of two regions (x and y). To simulate redundant interactions (ﬁrst row), each region receives a copy of the prediction (one-to-all) and for the synergy (second row), x and y receive early and late PE trials, respectively (all-to-one). This toy example illustrates that the local speciﬁcity is not the only factor determining the type of their interactions. We added the following result to the Appendix.

**Author response image 4. sa4fig4:** Local specificity does not fully determine the type of interactions. Within-area local encoding of PE using the mutual information (MI, in bits) for regions X and Y and between-area interaction information (II, in bits) leading to (A) redundant interactions and (B) synergistic interactions about the PE.

Regarding the information transfer results (Fig. 4), similar arguments hold and suggest that the prevalence is not the main factor explaining the arising transfer entropy between the anterior insula (aINS) and dorsolateral prefrontal cortex (dlPFC). Indeed, the lOFC has a strong local speciﬁcity for PPE, but the transfer entropy between the lOFC and aINS (or dlPFC) is shown in Fig. S7 does not show signiﬁcant differences in encoding between PPE and RPE.

Indeed, such transfer can only be found when there is a delay between the gamma activity of the two regions. In this example, the transfer entropy quantiﬁes the amount of information shared between the past activity of the aINS and the present activity of the dlPFC conditioned on the past activity of the dlPFC. The conditioning ensures that the present activity of the dlPFC is not only explained by its own past. Consequently, if both regions exhibit various prevalences toward reward and punishment but without delay (i.e. at the same timing), the transfer entropy would be null because of the conditioning. As a fact, between 10 to -20% of bipolar recordings show a selectivity to the reward PE (represented by a proportion of 40-60% of subjects, Fig.S4). However, the transfer entropy estimated from the aINS to the dlPFC across rewarding trials is ﬂat and clearly non-signiﬁcant. If the transfer entropy was a byproduct of the local speciﬁcity then we should observe an increase, which is not the case here.

**Reviewer #2:**
Summary:Reward and punishment learning have long been seen as emerging from separate networks of frontal and subcortical areas, often studied separately. Nevertheless, both systems are complimentary and distributed representations of rewards and punishments have been repeatedly observed within multiple areas. This raised the unsolved question of the possible mechanisms by which both systems might interact, which this manuscript went after. The authors skillfully leveraged intracranial recordings in epileptic patients performing a probabilistic learning task combined with model-based information theoretical analyses of gamma activities to reveal that information about reward and punishment was not only distributed across multiple prefrontal and insular regions, but that each system showed speciﬁc redundant interactions. The reward subsystem was characterized by redundant interactions between orbitofrontal and ventromedial prefrontal cortex, while the punishment subsystem relied on insular and dorsolateral redundant interactions. Finally, the authors revealed a way by which the two systems might interact, through synergistic interaction between ventromedial and dorsolateral prefrontal cortex.Strengths:Here, the authors performed an excellent reanalysis of a unique dataset using innovative approaches, pushing our understanding on the interaction at play between prefrontal and insular cortex regions during learning. Importantly, the description of the methods and results is truly made accessible, making it an excellent resource to the community.This manuscript goes beyond what is classically performed using intracranial EEG dataset, by not only reporting where a given information, like reward and punishment prediction errors, is represented but also by characterizing the functional interactions that might underlie such representations. The authors highlight the distributed nature of frontal cortex representations and propose new ways by which the information speciﬁcally ﬂows between nodes. This work is well placed to unify our understanding of the complementarity and speciﬁcity of the reward and punishment learning systems.

We thank the reviewer for the positive feedback. Please ﬁnd below our answers to the weaknesses raised by the reviewer.

Weaknesses:The conclusions of this paper are mostly supported by the data, but whether the ﬁndings are entirely generalizable would require further information/analyses.First, the authors found that prediction errors very quickly converge toward 0 (less than 10 trials) while subjects performed the task for sets of 96 trials. Considering all trials, and therefore having a non-uniform distribution of prediction errors, could potentially bias the various estimates the authors are extracting. Separating trials between learning (at the start of a set) and exploiting periods could prove that the observed functional interactions are speciﬁc to the learning stages, which would strengthen the results.

We thank the reviewer for this question. We would like to note that the probabilistic nature of the learning task does not allow a strict distinction between the exploration and exploitation phases. Indeed, the probability of obtaining the less rewarding outcome was 25% (i.e., for 0€ gain in the reward learning condition and -1€ loss in the punishment learning condition). Thus, participants tended to explore even during the last set of trials in each session. This is evident from the average learning curves shown in Fig. 1B of (Gueguen et al., 2021). Learning curves show rates of correct choice (75% chance of 1€ gain) in the reward condition (blue curves) and incorrect choice (75% chance of 1€ loss) in the punishment condition (red curves).

For what concerns the evolution of PEs, as reviewer #1 suggested, we added a new ﬁgure representing the single-subject estimates of the R/PPE (Fig S2). Here, the conﬁdence interval is obtained across all pairs of stimuli presented during the different sessions. We retrieved the general trend of the R/PPE converging toward zero around 10 trials. Both average reward and punishment prediction errors converge toward zero in approximately 10 trials, single-participant curves display large variability, also at the end of each session. As a reminder, the 96 trials represent the total number of trials for one session for the four pairs and the number of trials for each stimulus was only 24.

**Author response image 5. sa4fig5:** Single-subject estimation of predictions errors. Single-subject trial-wise reward PE (RPE - blue) and punishment PE (PPE - red), ± 95% confidence interval.

However, the convergence of the R/PPE is due to the average across the pairs of stimuli. In the ﬁgure below, we superimposed the estimated R/PPE, per pair of stimuli, for each subject. It becomes very clear that high values of PE can be reached, even for late trials. Therefore, we believe that the split into early/late trials because of the convergence of PE is far from being trivial.

**Author response image 6. sa4fig6:** Single-subject estimation of predictions errors per pair of stimuli. Single-subject trial-wise reward PE (RPE - blue) and punishment PE (PPE - red).

Consequently, nonzero PRE and PPE occur during the whole session and separating trials between learning (at the start of a set) and exploiting periods, as suggested by the reviewer, does not allow a strict dissociation between learning vs no-learning. Nevertheless, we tested the analysis proposed by the reviewer, at the local level. We splitted the 24 trials of each pair of stimuli into early, middle and late trials (8 trials each). We then reproduced Fig. 2 by computing the mutual information between the gamma activity and the R/PPE for subsets of trials: early (ﬁrst row) and late trials (second row). We retrieved signiﬁcant encoding of both R/PPE in the aINS, dlPFC and lOFC in both early and late trials. The vmPFC also showed signiﬁcant encoding of both during early trials. The only difference emerges in the late trials of the vmPFC where we found a strong encoding of the RPE only. It should also be noted that here since we are sub-selecting the trials, the statistical analyses are only performed using a third of the trials.

Taken together, the combination of high values of PE achieved even for late trials and the fact that most of the ﬁndings are reproduced even with a third of the trials does not justify the split into early and late trials here. Crucially, this latest analysis conﬁrms that the neural correlates of learning that we observed reﬂect PE signals rather than early versus late trials in the session.

**Author response image 7. sa4fig7:** MI between gamma activity and R/PPE using early and late trials. Time courses of MI estimated between the gamma power and both RPE (blue) and PPE (red) using either early or late trials (first and second row, respectively). Horizontal thick lines represent significant clusters of information (p<0.05, cluster-based correction, non-parametric randomization across epochs).

Importantly, it is unclear whether the results described are a common feature observed across subjects or the results of a minority of them. The authors should report and assess the reliability of each result across subjects. For example, the authors found RPE-speciﬁc interactions between vmPFC and lOFC, even though less than 10% of sites represent RPE or both RPE/PPE in lOFC. It is questionable whether such a low proportion of sites might come from different subjects, and therefore whether the interactions observed are truly observed in multiple subjects. The nature of the dataset obviously precludes from requiring all subjects to show all effects (given the known limits inherent to intracerebral recording in patients), but it should be proven that the effects were reproducibly seen across multiple subjects.

We thank the reviewer for this remark that has also been raised by the ﬁrst reviewer. This issue was raised by the ﬁrst reviewer. Indeed, we added a supplementary ﬁgure describing the number of unique subjects per brain region and per pair of brain regions (Fig. S1A) such as the number of bipolar derivations per region and per subject (Fig. S1B).

**Author response image 8. sa4fig8:** Single subject anatomical repartition. (A) Number of unique subject per brain region and per pair of brain regions. (B) Number of bipolar derivations per subject and per brain region.

Regarding the reproducibility of the results across subjects for the local analysis (Fig. 2), we also added the instantaneous proportion of subjects having at least one bipolar derivation showing a signiﬁcant encoding of the RPE and PPE (Fig. S4). We found a minimum proportion of approximately 30% of unique subjects having the effect in the lOFC and vmPFC, respectively with the RPE and PPE. On the other hand, both the aINS and dlPFC showed between 50 to 100% of the subjects having the effect. Therefore, local encoding of RPE and PPE was never represented by a single subject.

**Author response image 9. sa4fig9:** 

Similarly, we performed statistical analysis on interaction information at the single-subject level and counted the proportion of unique subjects having at least one pair of recordings with signiﬁcant redundant and synergistic interactions about the RPE and PPE (Fig. S5). Consistently with the results shown in Fig. 3, the proportions of signiﬁcant redundant and synergistic interactions are negative and positive, respectively. For the within-regions interactions, approximately 60% of the subjects with redundant interactions are about R/PPE in the aINS and about the PPE in the dlPFC and 40% about the RPE in the vmPFC. For the across-regions interactions, 60% of the subjects have redundant interactions between the aINS-dlPFC and dlPFC-lOFC about the PPE, and 30% have redundant interactions between lOFC-vmPFC about the RPE. Globally, we reproduced the main results shown in Fig. 3.

**Author response image 10. sa4fig10:** Inter-subjects reproducibility of redundant interactions about PE signals. Time-courses of proportion of subjects having at least one pair of bipolar derivation with a significant interaction information (p<0.05, cluster-based correction, non-parametric randomization across epochs) about the RPE (blue) or PPE (red). Data are aligned to the outcome presentation (vertical line at 0 seconds). Proportion of subjects with redundant (solid) and synergistic (dashed) interactions are respectively going downward and upward.

Finally, the timings of the observed interactions between areas preclude one of the authors' main conclusions. Speciﬁcally, the authors repeatedly concluded that the encoding of RPE/PPE signals are "emerging" from redundancy-dominated prefrontal-insular interactions. However, the between-region information and transfer entropy between vmPFC and lOFC for example is observed almost 500ms after the encoding of RPE/PPE in these regions, questioning how it could possibly lead to the encoding of RPE/PPE. It is also noteworthy that the two information measures, interaction information and transfer entropy, between these areas happened at non overlapping time windows, questioning the underlying mechanism of the communication at play (see Figures 3/4). As an aside, when assessing the direction of information ﬂow, the authors also found delays between pairs of signals peaking at 176ms, far beyond what would be expected for direct communication between nodes. Discussing this aspect might also be of importance as it raises the possibility of third-party involvement.

The local encoding of RPE in the vmPFC and lOFC is observed in a time interval ranging from approximately 0.2-0.4s to 1.2-1.4s after outcome presentation (blue bars in **Fig. 2A**). The encoding of RPE by interaction information covers a time interval from approximately 1.1s to 1.5s (blue bars in **Fig. 3B**, bottom right panel). Similarly, signiﬁcant TE modulations between the vmPFC and lOFC speciﬁc for PPE occur mainly in the 0.7s-1.1s range. Thus, it seems that the local encoding of PPE precedes the effects observed at the level of the neural interactions (II and TE). On the other hand, the modulations in MI, II and TE related to PPE co-occur in a time window from 0.2s to 0.7s after outcome presentation. Thus, we agree with the reviewer that a generic conclusion about the potential mechanisms relating the three levels of analysis cannot be drawn. We thus replaced the term “emerge from” by “occur with” from the manuscript which may be misinterpreted as hinting at a potential mechanism. We nevertheless concluded that the three levels of analysis (and phenomena) co-occur in time, thus hinting at a potential across-scales interaction that needs further study. Indeed, our study suggests that further work, beyond the scope of the current study, is required to better understand the interaction between scales.

Regarding the delay for the conditioning of the transfer entropy, the value of 176 ms reﬂects the delay at which we observed a maximum of transfer entropy. However, we did not use a single delay for conditioning, we used every possible delay between [116, 236] ms, as explained in the Method section. We would like to stress that transfer entropy is a directed metric of functional connectivity, and it can only be interpreted as quantifying statistical causality deﬁned in terms of predictacìbility according to the Wiener-Granger principle, as detailed in the methods. Thus, it cannot be interpreted in Pearl’s causal terms and as indexing any type of direct communication between nodes. This is a known limitation of the method, which has been stressed in past literature and that we believe does not need to be addressed here.

To account for this, we revised the discussion to make sure this issue is addressed in the following paragraph:

“Here, we quantiﬁed directional relationships between regions using the transfer entropy (Schreiber, 2000), which is a functional connectivity measure based on the Granger-Wiener causality principle. Tract tracing studies in the macaque have revealed strong interconnections between the lOFC and vmPFC in the macaque (Carmichael and Price, 1996; Öngür and Price, 2000). In humans, cortico-cortical anatomical connections have mainly been investigated using diffusion magnetic resonance imaging (dMRI). Several studies found strong probabilities of structural connectivity between the anterior insula with the orbitofrontal cortex and dorsolateral part of the prefrontal cortex (Cloutman et al., 2012; Ghaziri et al., 2017), and between the lOFC and vmPFC (Heather Hsu et al., 2020). In addition, the statistical dependency (e.g. coherence) between the LFP of distant areas could be potentially explained by direct anatomical connections (Schneider et al., 2021; Vinck et al., 2023). Taken together, the existence of an information transfer might rely on both direct or indirect structural connectivity. However, here we also reported differences of TE between rewarding and punishing trials given the same backbone anatomical connectivity (Fig. 4). [...] “

**Reviewer #3:**
Summary:The authors investigated that learning processes relied on distinct reward or punishment outcomes in probabilistic instrumental learning tasks were involved in functional interactions of two different cortico-cortical gamma-band modulations, suggesting that learning signals like reward or punishment prediction errors can be processed by two dominated interactions, such as areas lOFC-vmPFC and areas aINS-dlPFC, and later on integrated together in support of switching conditions between reward and punishment learning. By performing the well-known analyses of mutual information, interaction information, and transfer entropy, the conclusion was accomplished by identifying directional task information ﬂow between redundancy-dominated and synergy-dominated interactions. Also, this integral concept provided a unifying view to explain how functional distributed reward and/or punishment information were segregated and integrated across cortical areas.Strengths:The dataset used in this manuscript may come from previously published works (Gueguen et al., 2021) or from the same grant project due to the methods. Previous works have shown strong evidence about why gamma-band activities and those 4 areas are important. For further analyses, the current manuscript moved the ideas forward to examine how reward/punishment information transfer between recorded areas corresponding to the task conditions. The standard measurements such mutual information, interaction information, and transfer entropy showed time-series activities in the millisecond level and allowed us to learn the directional information ﬂow during a certain window. In addition, the diagram in Figure 6 summarized the results and proposed an integral concept with functional heterogeneities in cortical areas. These ﬁndings in this manuscript will support the ideas from human fMRI studies and add a new insight to electrophysiological studies with the non-human primates.

We thank the reviewer for the summary such as for highlighting the strengths. Please ﬁnd below our answers regarding the weaknesses of the manuscript.

Weaknesses:After reading through the manuscript, the term "non-selective" in the abstract confused me and I did not actually know what it meant and how it ﬁts the conclusion. If I learned the methods correctly, the 4 areas were studied in this manuscript because of their selective responses to the RPE and PPE signals (Figure 2). The redundancy- and synergy-dominated subsystems indicated that two areas shared similar and complementary information, respectively, due to the negative and positive value of interaction information (Page 6). For me, it doesn't mean they are "non-selective", especially in redundancy-dominated subsystem. I may miss something about how you calculate the mutual information or interaction information. Could you elaborate this and explain what the "non-selective" means?

In the study performed by Gueguen et al. in 2021, the authors used a general linear model (GLM) to link the gamma activity to both the reward and punishment prediction errors and they looked for differences between the two conditions. Here, we reproduced this analysis except that we used measures from the information theory (mutual information) that were able to capture linear and non-linear relationships (although monotonic) between the gamma activity and the prediction errors. The clusters we reported reﬂect signiﬁcant encoding of either the RPE and/or the PPE. From **Fig. 2**, it can be seen that the four regions have a gamma activity that is modulated according to both reward and punishment PE. We used the term “non-selective”, because the regions did not encode either one or the other, but various proportions of bipolar derivations encoding either one or both of them.

The directional information ﬂows identiﬁed in this manuscript were evidenced by the recording contacts of iEEG with levels of concurrent neural activities to the task conditions. However, are the conclusions well supported by the anatomical connections? Is it possible that the information was transferred to the target via another area? These questions may remain to be elucidated by using other approaches or animal models. It would be great to point this out here for further investigation.

We thank the reviewer for this interesting question. We added the following paragraph to the discussion to clarify the current limitations of the transfer entropy and the link with anatomical connections :

“Here, we quantiﬁed directional relationships between regions using the transfer entropy (Schreiber, 2000), which is a functional connectivity measure based on the Granger-Wiener causality principle. Tract tracing studies in the macaque have revealed strong interconnections between the lOFC and vmPFC in the macaque (Carmichael and Price, 1996; Öngür and Price,
2000). In humans, cortico-cortical anatomical connections have mainly been investigated using diffusion magnetic resonance imaging (dMRI). Several studies found strong probabilities of structural connectivity between the anterior insula with the orbitofrontal cortex and dorsolateral part of the prefrontal cortex (Cloutman et al., 2012; Ghaziri et al., 2017), and between the lOFC and vmPFC (Heather Hsu et al., 2020). In addition, the statistical dependency (e.g. coherence) between the LFP of distant areas could be potentially explained by direct anatomical connections (Schneider et al., 2021). Taken together, the existence of an information transfer might rely on both direct or indirect structural connectivity. However, here we also reported differences of TE between rewarding and punishing trials given the same backbone anatomical connectivity (**Fig. 4**). Our results are further supported by a recent study involving drug-resistant epileptic patients with resected insula who showed poorer performance than healthy controls in case of risky loss compared to risky gains (Von Siebenthal et al., 2017).”

References

Carmichael ST, Price J. 1996. Connectional networks within the orbital and medial prefrontal cortex of macaque monkeys. J Comp Neurol 371:179–207.

Cloutman LL, Binney RJ, Drakesmith M, Parker GJM, Lambon Ralph MA. 2012. The variation of function across the human insula mirrors its patterns of structural connectivity: Evidence from in vivo probabilistic tractography. NeuroImage 59:3514–3521. oi:10.1016/j.neuroimage.2011.11.016

Combrisson E, Allegra M, Basanisi R, Ince RAA, Giordano BL, Bastin J, Brovelli A. 2022. Group-level inference of information-based measures for the analyses of cognitive brain networks from neurophysiological data. NeuroImage 258:119347. doi:10.1016/j.neuroimage.2022.119347

Ghaziri J, Tucholka A, Girard G, Houde J-C, Boucher O, Gilbert G, Descoteaux M, Lippé S, Rainville P, Nguyen DK. 2017. The Corticocortical Structural Connectivity of the Human Insula. Cereb Cortex 27:1216–1228. doi:10.1093/cercor/bhv308

Gueguen MCM, Lopez-Persem A, Billeke P, Lachaux J-P, Rheims S, Kahane P, Minotti L, David O, Pessiglione M, Bastin J. 2021. Anatomical dissociation of intracerebral signals for reward and punishment prediction errors in humans. Nat Commun 12:3344. doi:10.1038/s41467-021-23704-w

Heather Hsu C-C, Rolls ET, Huang C-C, Chong ST, Zac Lo C-Y, Feng J, Lin C-P. 2020. Connections of the Human Orbitofrontal Cortex and Inferior Frontal Gyrus. Cereb Cortex 30:5830–5843. doi:10.1093/cercor/bhaa160

Lachaux J-P, Fonlupt P, Kahane P, Minotti L, Hoffmann D, Bertrand O, Baciu M. 2007. Relationship between task-related gamma oscillations and BOLD signal: new insights from combined fMRI and intracranial EEG. Hum Brain Mapp 28:1368–1375. doi:10.1002/hbm.20352

Mukamel R, Gelbard H, Arieli A, Hasson U, Fried I, Malach R. 2004. Coupling Between Neuronal Firing, Field Potentials, and fMRI in Human Auditory Cortex. Cereb Cortex 14:881.

Niessing J, Ebisch B, Schmidt KE, Niessing M, Singer W, Galuske RA. 2005. Hemodynamic signals correlate tightly with synchronized gamma oscillations. science 309:948–951.

Nir Y, Fisch L, Mukamel R, Gelbard-Sagiv H, Arieli A, Fried I, Malach R. 2007. Coupling between neuronal ﬁring rate, gamma LFP, and BOLD fMRI is related to interneuronal correlations. Curr Biol 17:1275–1285.

Öngür D, Price JL. 2000. The organization of networks within the orbital and medial prefrontal cortex of rats, monkeys and humans. Cereb Cortex 10:206–219.

Schneider M, Broggini AC, Dann B, Tzanou A, Uran C, Sheshadri S, Scherberger H, Vinck M. 2021. A mechanism for inter-areal coherence through communication based on connectivity and oscillatory power. Neuron 109:4050-4067.e12. doi:10.1016/j.neuron.2021.09.037

Schreiber T. 2000. Measuring information transfer. Phys Rev Lett 85:461.

Von Siebenthal Z, Boucher O, Rouleau I, Lassonde M, Lepore F, Nguyen DK. 2017. Decision-making impairments following insular and medial temporal lobe resection for drug-resistant epilepsy. Soc Cogn Affect Neurosci 12:128–137. doi:10.1093/scan/nsw152

**Recommendations for the authors**

**Reviewer #1**
(1) Overall, the writing of the manuscript is dense and makes it hard to follow the scientiﬁc logic and appreciate the key ﬁndings of the manuscript. I believe the manuscript would be accessible to a broader audience if the authors improved the writing and provided greater detail for their scientiﬁc questions, choice of analysis, and an explanation of their results in simpler terms.

We extensively modiﬁed the introduction to better describe the rationale and research question.

(2) In the introduction the authors state "we hypothesized that reward and punishment learning arise from complementary neural interactions between frontal cortex regions". This stated hypothesis arrives rather abruptly after a summary of the literature given that the literature summary does not directly inform their stated hypothesis. Put differently, the authors should explicitly state what the contradictions and/or gaps in the literature are, and what speciﬁc combinations of ﬁndings guide them to their hypothesis. When the authors state their hypothesis the reader is still left asking: why are the authors focusing on the frontal regions? What do the authors mean by complementary interactions? What speciﬁc evidence or contradiction in the literature led them to hypothesize that complementary interactions between frontal regions underlie reward and punishment learning?

We extensively modiﬁed the introduction and provided a clearer description of the brain circuits involved and the rationale for searching redundant and synergistic interactions between areas.

(3) Related to the above point: when the authors subsequently state "we tested whether redundancy- or synergy dominated interactions allow the emergence of collective brain networks differentially supporting reward and punishment learning", the Introduction (up to the point of this sentence) has not been written to explain the synergy vs. redundancy framework in the literature and how this framework comes into play to inform the authors' hypothesis on reward and punishment learning.

We extensively modiﬁed the introduction and provided a clearer description of redundant and synergistic interactions between areas.

(4) The explanation of redundancy vs synergy dominated brain networks itself is written densely and hard to follow. Furthermore, how this framework informs the question on the neural substrates of reward versus punishment learning is unclear. The authors should provide more precise statements on how and why redundancy vs. synergy comes into play in reward and punishment learning. Put differently, this redundancy vs. synergy framework is key for understanding the manuscript and the introduction is not written clearly enough to explain the framework and how it informs the authors' hypothesis and research questions on the neural substrates of reward vs. punishment learning.

Same as above

(5) While the choice of these four brain regions in context of reward and punishment learning does makes sense, the authors do not outline a clear scientiﬁc justiﬁcation as to why these regions were selected in relation to their question.

Same as above

(6) Could the authors explain why they used gamma band power (as opposed to or in addition to the lower frequency bands) to investigate MI. Relatedly, when the authors introduce MI analysis, it would be helpful to brieﬂy explain what this analysis measures and why it is relevant to address the question they are asking.

Please see our answer to the ﬁrst public comment. We added a paragraph to the discussion section to justify our choice of focusing on the gamma band only. We added the following sentence to the result section to justify our choice for using mutual-information:

The MI allowed us to detect both linear and non-linear relationships between the gamma activity and the PE

An extended explanation justifying our choice for the MI was already present in the method section.

(7) The authors state that "all regions displayed a local "probabilistic" encoding of prediction errors with temporal dynamics peaking around 500 ms after outcome presentation". It would be helpful for the reader if the authors spelled out what they mean by probabilistic in this context as the term can be interpreted in many different ways.

We agree with the reviewer that the term “probabilistic” can be interpreted in different ways. In the revised manuscript we changed “probabilistic” for “mixed”.

(8) The authors should include a brief description of how they compute RPE and PPE in the beginning of the relevant results section.

The explanation of how we estimated the PE is already present in the result section: “We estimated trial-wise prediction errors by ﬁtting a Q-learning model to behavioral data. Fitting the model consisted in adjusting the constant parameters to maximize the likelihood of observed choices etc.”

(9) It is unclear from the Methods whether the authors have taken any measures to address the likely difference in the number of electrodes across subjects. For example, it is likely that some subjects have 10 electrodes in vmPFC while others may have 20. In group analyses, if the data is simply averaged across all electrodes then each subject contributes a different number of data points to the analysis. Hence, a subject with more electrodes can bias the group average. A starting point would be to state the variation in number of electrodes across subjects per brain region. If this variation is rather small, then simple averaging across electrodes might be justiﬁed. If the variation is large then one idea would be to average data across electrodes within subjects prior to taking the group average or use a resampling approach where the minimum number of electrodes per brain area is subsampled.

We addressed this point in our public answers. As a reminder, the new version of the manuscript contains a ﬁgure showing the number of unique patients per region, the PE at per participant level together with local-encoding at the single participant level.

(10) One thing to consider is whether the reward and punishment in the task is symmetrical in valence. While 1$ increase and 1$ decrease is equivalent in magnitude, the psychological effect of the positive (vs. the negative) outcome may still be asymmetrical and the direction and magnitude of this asymmetry can vary across individuals. For instance, some subjects may be more sensitive to the reward (over punishment) while others are more sensitive to the punishment (over reward). In this scenario, it is possible that the differentiation observed in PPE versus RPE signals may arise from such psychological asymmetry rather than the intrinsic differences in how certain brain regions (and their interactions) may encode for reward vs punishment. Perhaps the authors can comment on this possibility, and/or conduct more in depth behavioral analysis to determine if certain subjects adjust their choice behavior faster in response to reward vs. punishment contexts.

While it could be possible that individuals display different sensitivities vis-à-vis positive and negative prediction errors (and, indeed, a vast body of human reinforcement learning literature seems to point in this direction; Palminteri & Lebreton, 2022), it is unclear to us how such differences would explain into the recruitment of anatomically distinct areas reward and punishment prediction errors. It is important to note here that our design partially orthogonalized positive and reward vs. negative and punishment PEs, because the neutral outcome can generate both positive and negative prediction errors, as a function of the learning context (reward-seeking and punishment avoidance). Back to the main question, for instance, Lefebvre et al (2017) investigated with fMRI the neural correlates of reward prediction errors only and found that inter-individual differences in learning rates for positive and negative prediction errors correlated with differences in the degree of striatal activation and not with the recruitment of different areas. To sum up, while we acknowledge that individuals may display different sensitivity to prediction errors (and reward magnitudes), we believe that such differences should translated in difference in the degree of activation of a given system (the reward systems vs the punishment one) rather than difference in neural system recruitment

(11) As summarized in Fig 6, the authors show that information transfer between aINS to dlPFC was PPE speciﬁc whereas the information transfer between vmPFC to lOFC was RPE speciﬁc. What is unclear is if these ﬁndings arise as an inevitable statistical byproduct of the fact that aINS has high PPE-speciﬁcity and that vmPFC has high RPE-speciﬁcity. In other words, it is possible that the analysis in Fig 3,4 are sensitive to fact that there is a larger proportion of electrodes with either PPE or RPE sensitivity in aINS and vmPFC respectively - and as such, the II analysis might reﬂect the dominant local encoding properties above and beyond reﬂecting the interactions between regions per se. Simply put, could the analysis in Fig 3B turn out in any other way given that there are more PPE speciﬁc electrodes in aINS and more RPE speciﬁc electrodes in vmPFC? Some options to address this question would be to limit the electrodes included in the analyses (in Fig 3B for example) so that each region has the same number of PPE and RPE speciﬁc electrodes included.

Please see the simulation we added to the revised manuscript (Fig. S10) demonstrating that synergistic interactions can emerge between regions with the same speciﬁcity.

Regarding the possibility that Fig. 3 and 4 are sensitive to the number of bipolar derivations being R/PPE speciﬁc, a counter-example is the vmPFC. The vmPFC has a few recordings speciﬁc to punishment (Fig. 2) in almost 30% of the subjects (Fig. S4). However, there is no II about the PPE between recordings of the vmPFC (Fig. 3). The same reasoning also holds for the lOFC. Therefore, the proportion of recordings being RPE or PPE-speciﬁc is not sufficient to determine the type of interactions.

(12) Related to the point above, what would the results presented in Fig 3A (and 3B) look like if the authors ran the analyses on RPE speciﬁc and PPE speciﬁc electrodes only. Is the vmPFC-vmPFC RPE effect in Fig 3A arising simply due to the high prevalence of RPE speciﬁc electrodes in vmPFC (as shown in Fig. 2)?

Please see our answer above.

**Reviewer #2:**
Regarding Figure 2A, the authors argued that their ﬁndings "globally reproduced their previously published ﬁndings" (from Gueguen et al, 2021). It is worth noting though that in their original analysis, both aINS and lOFC show differential effects (aINS showing greater punishment compared to reward, and the opposite for lOFC) compared to the current analysis. Although I would be akin to believe that the nonlinear approach used here might explain part of the differences (as the authors discussed), I am very wary of the other argument advanced: "the removal of iEEG sites contaminated with pathological activity". This raised some red ﬂags. Does that mean some of the conclusions observed in Gueguen et al (2021) are only the result of noise contamination, and therefore should be disregarded? The author might want to add a short supplementary ﬁgure using the same approach as in Gueguen (2021) but using the subset of contacts used here to comfort potential readers of the validity of their previous manuscript.

We appreciate the reviewer's concerns and understand the request for additional information. However, we would like to point out that the ﬁgure suggested by the reviewer is already present in the supplementary ﬁles of Gueguen et al. 2021 (see Fig. S2). The results of this study should not be disregarded, as the supplementary ﬁgure reproduces the results of the main text after excluding sites with pathological activity. Including or excluding sites contaminated with epileptic activity does not have a significant impact on the results, as analyses are performed at each time-stamp and across trials, and epileptic spikes are never aligned in time across trials.

That being said, there are some methodological differences between the two studies. To extract gamma power, Gueguen et al. ﬁltered and averaged 10 Hz sub-bands, while we used multi-tapers. Additionally, they used a temporal smoothing of 250 ms, while we used less smoothing. However, as explained in the main text, we used information-theoretical approaches to capture the statistical dependencies between gamma power and PE. Despite divergent methodologies, we obtained almost identical results.

The data and code supporting this manuscript should be made available. If raw data cannot be shared for ethical reasons, single-trial gamma activities should at least be provided. Regarding the code used to process the data, sharing it could increase the appeal (and use) of the methods applied.

We thank the reviewer for this suggestion. We added a section entitled “Code and data availability” and gave links to the scripts, notebooks and preprocessed data.